# CXAD: Contrastive Explanations for Anomaly Detection: Algorithms, Complexity Results and Experiments

**Ian Davidson**                                                                     *indavidson@ucdavis.edu*
*Department of Computer Science*
*University of California Davis*

**Nicolás Kennedy**                                                                  *nbkennedy@ucdavis.edu*
*Department of Computer Science*
*University of California Davis*

**S. S. Ravi**                                                                       *ssravi0@gmail.com*
*Biocomplexity Institute*
*University of Virginia and*
*Department of Computer Science*
*University at Albany – SUNY*

**Reviewed on OpenReview:** *https://openreview.net/forum?id=Tnwci2kLna&nesting=2&sort=date-desc*

## Abstract

Anomaly/Outlier detection (AD/OD) is often used in controversial applications to detect unusual behavior which is then further investigated or policed. This means an explanation of why something was predicted as an anomaly is desirable not only for individuals but also for the general population and policy-makers. However, existing explainable AI (XAI) methods are not well suited for Explainable Anomaly detection (XAD). In particular, most XAI methods provide instance-level explanations, whereas a model/global-level explanation is desirable for a complete understanding of the definition of normality or abnormality used by an AD algorithm. Further, existing XAI methods try to explain an algorithm's behavior by finding an explanation of why an instance belongs to a category. However, by definition, anomalies/outliers are chosen because they are different from the normal instances. We propose a new style of model agnostic explanation, called contrastive explanation, that is designed specifically for AD algorithms which use semantic tags to create explanations. It addresses the novel challenge of providing a model-agnostic and global-level explanation by finding contrasts between the outlier group of instances and the normal group. We propose three formulations: (i) Contrastive Explanation, (ii) Strongly Contrastive Explanation, and (iii) Multiple Strong Contrastive Explanations. The last formulation is specifically for the case where a given dataset is believed to have many types of anomalies. For the first two formulations, we show the underlying problem is in the computational class **P** by presenting linear and polynomial time exact algorithms. We show that the last formulation is computationally intractable, and we use an integer linear program for that version to generate experimental results. We demonstrate our work on several data sets such as the CelebA image data set, the HateXplain language data set, and the COMPAS dataset on fairness. These data sets are chosen as their ground truth explanations are clear or well-known.

## 1 Introduction

Anomaly[1] detection (AD) involves identifying unusual instances in a dataset. It is a central part of artificial intelligence (AI) and perhaps the most controversial given that it is employed in high-impact applications that

---

[1] We will use the terms anomaly and outlier interchangeably though there are nuanced differences.

typically require intervention, policing, and investigation. For example, it is used in social networks (Savage et al., 2014) to identify fake accounts and hate speech, by financial companies to identify fraudulent credit card transactions (Ahmed et al., 2016) and by the government to identify fraudulent benefit claims (van Capelleveen et al., 2016). Being typically unsupervised, the need for explanation is even more necessary than for supervised learning for a multitude of reasons including fairness (e.g., is a certain ethnic group being excluded from social networks?), transparency to the individual (e.g., why were certain credit card transactions declined?) and elucidation to policymakers (e.g., how are people committing fraudulent benefit claims?).

**Previous XAI Work and its Limitations.** Though Explainable AI (XAI) (see e.g., Adadi & Berrada (2018); Proc. XAI-2017 Workshop; Proc. XAI-2018 Workshop; Gunning (2017); Dosilovic et al. (2018); Zhang & Chen (2018); Ming (2017); Zheng et al. (2018)) methods have made tremendous strides, they are not particularly suitable for Explainable Anomaly Detection (XAD) for several reasons we now describe. Existing two-class XAI methods for supervised learning such as LIME (Bodria et al., 2023) identify which parts of an instance are responsible for its classification ("Why did you do that?"). That is, these are inclusive and instance-level explanations that state what properties an instance has to belong to a class. However, these styles of explanations are not entirely suitable for AD as a definition of an anomaly is that it deviates from the population; hence, this requires the normal (non-outlier) points to be part of the explanatory mechanism. Thus, our focus is on contrastive explanations that differentiate anomalies form normal instances and vice-versa.

Further, a natural higher-level question for policy designers and the general public is: how does an algorithm define normality and anomalousness? As model-level explanations in terms of the feature space are challenging for deep learners and are data modality-specific, we formulate a model agnostic approach using human interpretable tags. These tags can be created by humans or automatically for a variety of sources as shown in Table 1. Our method can also be employed when the features used to carry out AD are interpretable as the features can also serve as the tags; we demonstrate this in our experiments on the COMPAS data set. Model/global style explanations are not well studied for supervised or unsupervised learning, for example there is just a few papers for clustering (Davidson et al., 2018; Sambaturu et al., 2020). However, these works are not suitable for XAD as they generate explanations in terms of the properties of the instances within each class/cluster and we wish to understand better why the group of outliers is identified in terms of not only what they share in common with each other, but also what they do not share in common with the normal points. The work on XAD is relatively new but an excellent survey article recently was published (Li et al., 2024). We compare our work with existing XAD work in Section 6 after we present our methods and experimental results.

**Contributions.**

- We tackle the understudied problem of XAD which requires a different style of explanation compared to existing XAI methods in classification and clustering. We study XAD using a contrastive mechanism which is consistent with how humans explain to each other (Miller, 2021); see the discussion in Section 6 ("Related XAD Work").
- We formulate a model agnostic contrastive form of model/global-level XAD through a variety of optimization problems by interpreting them as computations on an appropriate bipartite graph. By drawing connections to the literature in theoretical computer science, we show that XAD can be viewed as a Knapsack style problem. Model agnosticism is particularly important for XAD as there are many popular but different styles of algorithms such as those that measure properties rather than optimizing a function.
- We show that for two of our formulations, polynomial time exact algorithms (including a simple to implement dynamic programming algorithm) can be developed allowing them to scale to very large data sets. For more complex multi-part explanations, we prove that the corresponding problem is **NP**-hard but construct an integer linear program (ILP) that is readily solvable with large-scale solvers such as Gurobi.
- Finally, we demonstrate the effectiveness of our methods on multiple modalities of data where ground truth explanations are known to exist: Images (Celebrity A Faces), Text (HateXPlain), and Databases (COMPAS for Fairness).

**Organization.** We begin with an overview of our method and then formalize our problem definitions. Next, we present efficient algorithms for the first two formulations and a complexity result for the third. Finally, we discuss our experimental results and related work. We close with a summary and some concluding remarks.

## 2 Overview of Our Approach

**Core Idea for Constrastive Explanation.** We can explain our contrastive explanation approach by visualizing a bipartite graph but with three types of nodes shown in Figure 1: outlier nodes representing outlier instances, normal nodes representing inliers and tag nodes, each representing a semantic property. Note this is not strictly a tri-partite graph as there can be no edges between the nodes representing the outlier (left) and normal (right) instances. Instead, outlier and normal instance nodes can only have edges to the tag nodes. For example, normal and outlier nodes for facial image data are connected to only to the tags describing the image (e.g., race, hair color, gender). As mentioned in the introduction, the tags can be part of the features used to perform anomaly detection as we have done in our experiments with the COMPAS (fairness) database.

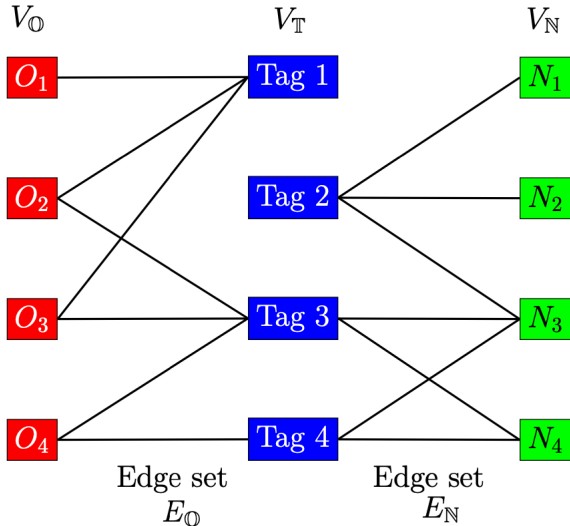

Figure 1: The underlying graph on which computations are performed to find explanations. Node sets $V_{\mathbb{O}}$, $V_{\mathbb{N}}$ and $V_{\mathbb{T}}$ denote the sets of outliers, normal instances and the set of all tags respectively. An edge between a tag and an instance indicates that the instance has that tag. The symbols $E_{\mathbb{O}}$ and $E_{\mathbb{N}}$ represent respectively the set of edges between outliers and tags and that between the normal instances and tags. Note that solutions to our problem formulations are one or more subsets (denoted by $Y$) of $V_{\mathbb{T}}$.

The underlying computation is to choose a subset of tags that together form an explanation/coverage for many outliers but not explain/cover the normal points. This is so as to create a contrastive explanation. Explanation quality can be measured in several ways and we choose a measurement based on the number of edges as it can give rise to efficient computations. Computations based on coverage typically are computationally intractable as they can model versions of the minimum set cover problem; see (Garey & Johnson, 1979) for a definition of this problem.

The outlier and normal instances can be switched and the computation repeated to generate an explanation for normal points; for the remainder of this section, we focus on explaining outliers for simplicity. We formalize these styles of computation by borrowing analogies from packing problems in theoretical computer science. We present informal definitions and formulations below to provide an overview of our approach. Formal definitions and precise problem formulations appear in Section 3.

**Definition 2.1.** ***Profit and Weight of Tags.*** *Each tag is considered an item to pack into an outlier explanation and its profit is determined by the number of edges incident on it from the outlier instances,*

*whilst its weight is determined by the edges incident on it from the normal instances. (As mentioned earlier, we can switch the role of the outlier and normal instances to get an explanation for normal instances.)*

For example, in Figure 1, `Tag 1` has a profit of 3 and weight of 0 and would be a good tag to describe outliers, where as `Tag 2` (profit of 0 and weight 3) would not.

In our first formulation, we wish to choose a subset of tags to maximize the number of edges covered by these tags for outlier points less the number of edges covered by normal points. (In Section 3, we will formally define this difference as the **utility** of the subset of tags.) That is, the objective of this formulation is to optimize profit minus weight. Each tag then naturally has a measure of utility which is simply the number of tags incident on the outlier points less the number of tags incident on the normal points. In Figure 1, to explain the outlier instances shown, one would choose the subset {`Tag 1`, `Tag 3`} of tags as all other tags have a non-positive utility when explaining the outlier set. This idea leads to a simple linear time algorithm and we also consider a variation that maximizes the utility under a budget constraint on the number of tags chosen for the explanation. Formally, using the terminology in Figure 1, our first formulation is as follows. Let $E_{\mathbb{O}}^Y$ and $E_{\mathbb{N}}^Y$ be the subset of edges incident on the outlier and normal instances (respectively) when $Y$ is the chosen subset of tags for the explanation.

**Formulation #1: Contrastive Explanation.** Here we select a subset of tags to optimize the quantity profit - weight: $\operatorname{argmax}_{Y \subseteq \mathbb{U}} |E_{\mathbb{O}}^Y| - |E_{\mathbb{N}}^Y|$. A variant will explore limiting the explanation complexity to $k$ tags. We formalize these as Description of Maximum Utility (DMU) and Budgeted Description of Maximum Utility (BDMU) problems in Section 3.2 and develop an exact linear time algorithm for them (see Algorithm 1 and Theorem 4.1).

The explanation provided by the above formulation for the problem in Figure 1 is {`Tag 1`, `Tag 3`} which does explain the outlier points, but it is not strongly contrastive as it explains several normal points as well. In some circumstances we wish to generate strongly contrastive explanations. Our second formulation addresses this by extending our packing analogy to the Knapsack problem (Garey & Johnson, 1979) with the knapsack of chosen tags representing the tags that are most explanatory. This allows us to choose a subset of tags to maximize profit (i.e., the number of edges incident on outlier points) whilst upper bounding ($\beta$) the weight (i.e., the number of edges incident on normal points). This effectively changes the computation to choosing the most contrastive tags between the two groups. In Figure 1 to explain the outlier instances with $\beta = 1$, one would choose just `Tag 1`. In general, the Knapsack problem is computationally intractable (Garey & Johnson, 1979). However, the version of Knapsack arising in the context of contrastive explanation involves numbers whose values are bounded by polynomial functions of the problem size. This restriction leads to an efficient exact algorithm for the explanation problem through dynamic programming. Formally, our second formulation is as follows.

**Formulation #2: Strongly Contrastive Explanation.** Here we select tags to maximize utility whilst upper bounding weight: $\operatorname{argmax}_{Y \subseteq \mathbb{U}} |E_{\mathbb{O}}^Y|$ s.t. $|E_{\mathbb{N}}^Y| \le \beta$. In Section 3.2, we formalize this as a the Description of Maximum Profit under a Weight Constraint (DMP-WC) problem, which is a Knapsack problem where the items are tags and the profit is the number of edges incident on $\mathbb{O}$ and the weight is the number of edges incident on $\mathbb{N}$. We present an exact dynamic programming-based polynomial time algorithm for this problem (see Algorithm 2 and Theorem 4.2).

Finally, our third formulation removes the limitation of generating just one explanation; instead, it allows many explanations by formulating the problem as a multi-knapsack problem, with each knapsack being an explanation. This naturally allows not only the generation of more complex (multi-cause) explanations but also allows addressing challenging settings. One such setting we shall study is to compare different OD algorithms to understand how they are different from each other in terms of results. Though the problem resulting from this formulation is **NP**-hard as we shall prove, it serves the purposes of exploring complex explanations. A definition of this formulation is as follows.

**Formulation #3: Multiple Strongly Contrastive Explanations.** Here we extend the previous formulation, but rather than just one explanation/knapsack we allow multiple knapsacks: $\operatorname{argmax}_{Y_1 \subseteq \mathbb{U}, \ldots, Y_\sigma \subseteq \mathbb{U}} \sum_i |E_{\mathbb{O}}^{Y_i}|$ s.t. $|E_{\mathbb{N}}^{Y_i}| \le \beta$, $1 \le i \le \sigma$ and $Y_i \cap Y_j = \emptyset$, $\forall i, j \; i \ne j$. The previous formulations assume one explanation for all instances in a group. However, as there may be different

types of outliers, we formulate a multi-knapsack version. We call this version the Multiple Descriptions of Maximum Profit Under a Weight Constraint (MDMP-WC) problem in Section 3.2 and show that it is computationally intractable; see Theorem 4.3. For our experiments, we use an integer linear program (ILP) to obtain an exact solution to this version.

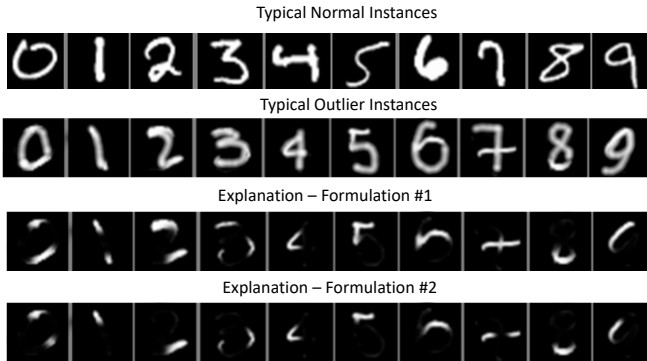

Figure 2: An illustrative application of our first two formulations to explain the outliers in the MNIST dataset. For each of the ten digit classes we created a separate outlier detection problem (shown as columns) using a basic autoencoder. For each outlier detection problem, we show a typical normal and outlier instance and the explanations found by our first two methods. Our first formulation finds all contrastive pixels which occur more often in the outlier group than in the normal group. Our second formulations finds the most contrastive pixels.

**Illustrative Example Using MNIST.** We provide an illustrative example of the first two formulations on the MNIST digit data set. For each of the ten digits, we created a different outlier detection problem and generated an explanation for each. In this data set, the tags are the pixels of the digits themselves; hence, we are explaining outliers using the features. Our third explanatory approach is for more complex explanations which do not occur in this simple data set. We apply a basic auto-encoder to determine outliers based on their reconstruction error for each digit class separately. We then apply our method separately to the top 5% most anomalous instances for each digit type. The remaining digit's instances are deemed normal which means some instances deemed normal are still quite unusual. Some typical normal points are shown in the first row of Figure 2 and typical outliers are in the second row. A brief description of the explanations found by our formulations are provided in the caption of Figure 2.

**Tag Generation, Assumptions and Limitations.** Our three formulations represent successive relaxations in that they naturally trade-off explaining all instances with explanation complexity. As such, they do not have any strong assumptions beyond requiring tags that are typically representative of the instances. A legitimate question is how to determine if this assumption is violated and more generally measure explanation quality. Contrastive explanation can naturally identify if the tags are not correlated well with the outcomes of an AD algorithm; that is, there does not exist a strongly contrastive explanation. For example, in Table 2 (Section 5), if the two percentages reported for each tag found in the explanation were approximately the same (or not significantly different), that would indicate a non-constrastive (and poor) explanation. Our method requires the existence of tags which could of course be generated by humans, however, Table 1 indicates other automated sources of tags.

Finally, we note that like just about all XAI methods (to our knowledge), our method does not explain the thought process of the deep learner, a topic known as meta-cognition in the cognitive science literature (Lai, 2011) which is gaining attention in AI (Walker et al., 2025; Caro et al., 2014).

Table 1: Some sources of tags for different instance modalities.

| Modality | Tag Creation Methods |
|---|---|
| People in Images | Commercial software such as Google Photos can tag images with the names of people within them. The associated files can be downloaded for analysis. For example, EXIF files list the Tags/DateTimeOriginal used in Google Photos and IPTC files list the Tags/Caption-Abstract used for the Description in Google Photos. |
| Objects in Images | Many image taggers exist (e.g., blip2, kohya, wd14-swinv2-v2). Further, there are tag editors. The creation of StableDiffusion methods for auto-captioning of images also provides methods to generate tags. |
| Emails, articles, documents | LLMs in principle have allowed the possibility of tagging documents with keywords but the concern was how useful these tags would be for specific domains. However, recent work has shown how to take general LLMs and make them task specific taggers of documents, emails etc. (Shen et al., 2024). |
| Graphs | The ability to tag social network users has been in practice for well over a decade and much meta-data on users is available (Gupta et al., 2010). |

## 3 Formal Problem Definitions

Here, we provide more rigorous versions of our previous problem formulations. The rigorous definitions enable us to develop efficient algorithms for some versions and prove computational intractability results for other versions.

### 3.1 Basic Definitions

We assume that the dataset $\mathbb{S}$ has been partitioned into two blocks, namely the **normal** set $\mathbb{N}$ and the **outlier** set $\mathbb{O}$. We are also given a (finite) universal tag set $\mathbb{U}$. For each instance $s_i \in \mathbb{S}$, we have an associated subset $T_i \subseteq \mathbb{U}$ of tags. Each tag $\tau \in T_i$ is considered an explanation of instance $s_i$; formally, we say that each tag $\tau \in T_i$ **covers** the instance $s_i$.

To formulate problems and develop algorithmic results, we use an undirected bipartite graph $G(V_{\mathbb{N}+\mathbb{O}}, V_{\mathbb{T}}, E)$ to represent the instances, tags and the covering relationship between the tags and instances. The node set $V_{\mathbb{N}+\mathbb{O}}$ of this graph consists of two (disjoint) subsets, namely $V_{\mathbb{N}}$ and $V_{\mathbb{O}}$. Each node in $V_{\mathbb{N}}$ represents a normal instance while each node in $V_{\mathbb{O}}$ represents an outlier instance. Each node in $V_{\mathbb{T}}$ represents a tag. Each edge $\{x, y\}$ of $G$, where $x \in V_{\mathbb{N}+\mathbb{O}}$ and $y \in V_{\mathbb{T}}$, indicates that the instance represented by $x$ is covered by the tag represented by $y$. Even though $G$ is a bipartite graph, it will be convenient to visualize it using three pairwise disjoint node sets as in Figure 1 so that one can readily recognize the sets of normal and outlier instances covered by each tag.

We partition the edge set $E$ into two subsets, namely $E_{\mathbb{N}}$ and $E_{\mathbb{O}}$. Here, $E_{\mathbb{N}}$ denotes the set of edges between $V_{\mathbb{N}}$ and $V_{\mathbb{T}}$ while $E_{\mathbb{O}}$ denotes the set of edges between $V_{\mathbb{O}}$ and $V_{\mathbb{T}}$. The bipartite graph $G$ representation is sufficient for formulating our optimization problems and for developing our analytical results. So, we will refer to the nodes in $V_{\mathbb{N}}$ themselves as the normal instances, the ones in $V_{\mathbb{O}}$ as the outlier instances and those in $V_{\mathbb{T}}$ as the tags.

For a subset $Y \subseteq V_{\mathbb{T}}$ of tags, let $\mathbb{N}(Y)$ and $\mathbb{O}(Y)$ denote respectively the subset of $V_{\mathbb{N}}$ and $V_{\mathbb{O}}$ covered by $Y$. Thus, in the bipartite graph $G$, each node in $\mathbb{N}(Y) \cup \mathbb{O}(Y)$ has an edge to at least one node in $Y$. For notational simplicity, when $Y$ consists of a single node, say $b$, we will use $\mathbb{N}(b)$ and $\mathbb{O}(b)$ (instead of $\mathbb{N}(\{b\})$ and $\mathbb{O}(\{b\})$). For a subset of tags $Y$, let $E_{\mathbb{N}}^Y$ denote the set of edges between the nodes in $Y$ and $V_{\mathbb{N}}$; further, let $E_{\mathbb{O}}^Y$ denote the set of edges between the nodes in $Y$ and $V_{\mathbb{O}}$. Here again, when $Y$ consists of a single node $b$, we will use the simpler notation $E_{\mathbb{N}}^b$ and $E_{\mathbb{O}}^b$.

**Optimization Objectives.** For simplicity, our problem definitions assume that we are focusing on an explanation for the outlier points. These definitions can readily flipped to explain normal instances as we have done in our experiments. For any tag node $y$, the **profit** of $y$, denoted by $p(y)$, is the number of outlier

instances covered by $y$; that is, $p(y) = |\mathbb{O}(y)|$. The **weight** of any tag $y$, denoted by $w(y)$, is the number of normal instances covered by $y$; that is, $w(y) = |\mathbb{N}(y)|$. For any tag node $y$, the **utility** of $y$, denoted by $\mu(y)$, is given by $p(y) - w(y)$; thus, the utility of a tag is the number of outlier instances covered by $y$ minus the number of normal instances covered by $y$. In our first set of formulations, it is desirable to choose tags with *positive* utility since each such tag explains more outlier instances than normal instances.

We can extend the definitions of utility, weight and profit to subsets of tags as follows. For any subset $Y$ of tags, (i) the profit $p(Y)$ is given by $\sum_{y \in Y} p(y)$, (ii) the weight $w(Y)$ of $Y$ is given by $\sum_{y \in Y} w(y)$, and (iii) the utility $\mu(Y)$ of $Y$ is given by $\sum_{y \in Y} \mu(y)$. The following simple observation relates these extensions to edge subsets of the graph $G$.

**Observation 3.1.** *For any subset $Y$ of tags, the following equations hold: (i) $p(Y) = |E_{\mathbb{O}}^{Y}|$, (ii) $w(Y) = |E_{\mathbb{N}}^{Y}|$ and (iii) $\mu(Y) = |E_{\mathbb{O}}^{Y}| - |E_{\mathbb{N}}^{Y}|$.*

This observation is useful in pointing out that the rigorous specifications of the combinatorial problems presented below are equivalent to the formulations mentioned in Section 2.

### 3.2 Main Problem Formulations

We can now formally specify the optimization problems considered in this paper. Problems (a) and (b) formulated below correspond to two versions of Formulation #1 in Section 2. The problem names have been chosen to explicitly indicate the optimization objective.

**(a) Description of Maximum Utility** (DMU) and **(b) Budgeted Description of Maximum Utility** (BDMU)

Given: Bipartite graph $G(V_{\mathbb{N}+\mathbb{O}}, V_{\mathbb{T}}, E)$ that represents normal instances, outlier instances, tags and the covering relationship between tags and instances.

Required: (a) For the DMU problem, the goal is to find a subset $Y^*$ of tags such that the utility $\mu(Y^*)$ of $Y^*$ is a maximum among all subsets of tags. (b) For the BDMU problem, the input also includes an integer budget $k \leq |V_{\mathbb{T}}|$, and the goal is to find a subset $Y^* \subseteq V_{\mathbb{T}}$ such that $|Y^*| \leq k$ and the utility $\mu(Y^*)$ of $Y^*$ is a maximum among all subsets of tags of size at most $k$.

Using Observation 3.1, it is easy to see that the goal of the DMU problem is to find a subset $Y^*$ of tags to maximize the quantity $|E_{\mathbb{O}}^{Y^*}| - |E_{\mathbb{N}}^{Y^*}|$. This version was given as Formulation #1 in Section 2. The optimization goal of the BDMU problem is also the same as that of DMU except that $Y^*$ needs to satisfy an additional budget constraint, namely $|Y^*| \leq k$.

Our next problem corresponds to Formulation #2 of Section 2.

**(c) Description of Maximum Profit Under a Weight Constraint** (DMP-WC)

Given: Bipartite graph $G(V_{\mathbb{N}+\mathbb{O}}, V_{\mathbb{T}}, E)$ that represents normal instances, outlier instances, tags and the covering relationship between tags and instances; a positive integer $W$.

Required: A subset $Y^* \subseteq V_{\mathbb{T}}$ such that the weight $w(Y^*)$ is at most $W$ and the profit $p(Y^*)$ is a maximum among all subsets of tags satisfying the weight constraint.

Using Observation 3.1, it is seen that the goal of the DMP-WC problem is to find a subset $Y^*$ of tags to maximize the quantity $|E_{\mathbb{O}}^{Y^*}|$ subject to the constraint $|E_{\mathbb{N}}^{Y^*}| \leq W$. This version was given as Formulation #2 in Section 2.

Our final problem corresponds to Formulation #3 of Section 2.

**(d) Multiple Descriptions of Maximum Profit Under a Weight Constraint** (MDMP-WC)

Given: Bipartite graph $G(V_{\mathbb{N}+\mathbb{O}}, V_{\mathbb{T}}, E)$ that represents normal instances, outlier instances, tags and the covering relationship between tags and instances; positive integers $\sigma$ and $W$.

Required: Pairwise disjoint subsets $Y_1$, $Y_2$, ..., $Y_\sigma$ of $V_\mathbb{T}$ such that for $1 \le j \le \sigma$, the weight $w(Y_j)$ is at most $W$ and the total profit $\sum_{j=1}^{\sigma} p(Y_j)$ is a maximum among all collections of $\sigma$ pairwise disjoint subsets of tags that satisfy the weight constraint.

Using the relationship between the profit and weight of a set $Y$ of tags and the edges in $G$, it is seen that the goal of the DMP-WC problem is to find pairwise disjoint subsets $Y_1$, $Y_2$, ..., $Y_\sigma$ of $V_\mathbb{T}$ to maximize the quantity $\sum_{i=1}^{\sigma} |E_\mathbb{O}^{Y^*}|$, subject to the constraint $|E_\mathbb{N}^{Y_i}| \le W$, $1 \le i \le \sigma$. This is Formulation #3 in Section 2.

This completes the formal definitions of the problems considered in our paper. In the next section, we show that Problems (a), (b) and (c) formulated above can be solved efficiently while Problem (d) is computationally intractable.

**A note about experiments.** In presenting experimental results, we normalize values of parameters by constants to make the computational results more meaningful and comparable across datasets of different sizes. For example, the experiments for the DMU problem (Formulation #1) choose a subset of tags $Y$ to maximize the quantity $\frac{|E_\mathbb{O}^Y|}{|E_\mathbb{O}|} - \frac{|E_\mathbb{N}^Y|}{|E_\mathbb{N}|}$, where $|E_\mathbb{O}|$ represents the total number of edges between $\mathbb{O}$ and $V_\mathbb{T}$ and $|E_\mathbb{N}|$ represents the total number of edges between $\mathbb{O}$ and $V_\mathbb{T}$. The objectives in Formulations #2 and #3 are scaled in a similar fashion.

## 4 Analytical Results

### 4.1 Algorithms for Maximizing Utility

Here, we present efficient algorithms for the first two versions of Formulation#1, namely DMU and BDMU, formulated in Section 3. In both cases, the goal is find a subset of tags to maximize the utility. The correctness of our algorithms is due to the following lemma.

**Lemma 4.1.** *An optimal solution to the DMU problem consists of all tags whose utility values are $> 0$.*

**Proof:** Let $Y^*$ be an optimal solution to the the given DMU problem. By definition, the optimal utility $\mu(Y^*)$ is given by $\mu(Y^*) = \sum_{y \in Y^*} \mu(y)$. We may assume without loss of generality that $Y^*$ doesn't contain any tag $y$ with $\mu(y) = 0$ since deleting such a tag doesn't change the total utility. If $Y^*$ doesn't contain a tag $y$ with $\mu(y) > 0$, then *adding* that tag to $Y^*$ increases the total utility, contradicting the optimality of $Y^*$. Thus, $Y^*$ must contain all tags whose utility values are $> 0$. Further, if $Y^*$ includes any tag $y$ with $\mu(y) < 0$, then *deleting* $y$ from $Y^*$ increases the total utility, again contradicting the optimality of $Y^*$. Therefore, the set $Y^*$ containing each tag with *strictly* positive utility is an optimal solution.

The above lemma suggests the simple algorithm for the DMU problem shown in Algorithm 1. The following simple extension of this algorithm also solves the BDMU problem, where the goal is to choose a subset $Y$ of at most $k$ tags with maximum utility. Once we have computed the utilities of all the tags, we only need to choose the top $k$ tags when tags are listed in non-increasing order of utilities. (If the resulting subset includes tags whose utility values are $\le 0$, they can be discarded.) As will be pointed out in the proof of Theorem 4.1 below, choosing the top $k$ tags can be done *without* sorting the tags in the order of their utilities.

**Theorem 4.1.** *Problems DMU and BDMU can be solved in time $O(|V_{\mathbb{N}+\mathbb{O}}| + |V_\mathbb{T}| + |E|)$, which is linear in the size of the graph $G$.*

**Proof:** The correctness of our algorithms for DMU and BDMU follows from Lemma 4.1 and the discussion following our proof of the lemma. To estimate their running times, we first consider Algorithm 1 for the DMU problem. The time to read the graph $G$ is $O(|V_{\mathbb{N}+\mathbb{O}}| + |V_\mathbb{T}| + |E|)$. For any tag $y$, let degree($y$) denote the degree of $y$ in $G$. The utility $\mu(y)$ of $y$ can be computed in $O(\text{degree}(y))$ time by counting the number of edges between $y$ and the nodes in $V_{\mathbb{N}+\mathbb{O}}$. The time used to check whether $y$ is added to $Y^*$ is $O(1)$. Thus, the total time used in the **for** loop of the algorithm is $O\left(\sum_{u \in V_\mathbb{T}} \text{degree}(u)\right) = O(|E|)$ since by a well known graph theoretic fact, the total degree of all the nodes in $V_\mathbb{T}$ is simply twice the number of edges in $G$ (West, 2001). Therefore, the running time of Algorithm 1 $O(|V_{\mathbb{N}+\mathbb{O}}| + |V_\mathbb{T}| + |E|)$.

---

**Algorithm 1:** An Algorithm for the Description of Maximum Utility (DMU) problem

---

**Input** : A bipartite graph $G(V_{\mathbb{N}+\mathbb{O}}, V_{\mathbb{T}}, E)$ representing the normal instances, outlier instances and tags as discussed in Section 3.

**Output:** A subset $Y^* \subseteq V_{\mathbb{T}}$ of tags with maximum utility.

**1** Initialize $Y^*$ to the empty set.
**2 for** each tag $y \in V_{\mathbb{T}}$ **do**
**3** | Compute the utility $\mu(y)$ of $y$.
**4** | **if** $\mu(y) > 0$ **then**
**5** | | Add $y$ to $Y^*$.
**6** | **end**
**7 end**

**8** Output $Y^*$.

---

Now, consider the extended algorithm for the BDMU problem. We need to choose the top-$k$ tags with the largest utility. As before, we can compute the utilities of all the $|V_{\mathbb{T}}|$ tags in $O(|V_{\mathbb{N}+\mathbb{O}}| + |V_{\mathbb{T}}| + |E|)$ time. It is well known that given an array of $n$ values and an integer $k \leq n$, the $k^{\text{th}}$ largest value in the array can be found in $O(n)$ time, without sorting the array (Cormen et al., 2009). Thus, we can find the top-$k$ tags with respect to utility in $O(|V_{\mathbb{T}}|)$ time. In other words, the running time of our algorithm for BDMU problem is also $O(|V_{\mathbb{N}+\mathbb{O}}| + |V_{\mathbb{T}}| + |E|)$.

### 4.2 Maximizing Profit Under Weight Constraint

In this section, we consider the DMP-WC problem, where the goal is to find a subset $Y^*$ of tags to maximize the total profit under a budget constraint on the total weight. We present a polynomial time algorithm for the problem by observing that this is a restricted version of the Knapsack problem (Cormen et al., 2009). We begin with a definition of the Knapsack problem.

**Knapsack Problem**

Given: A set $A = \{a_1, a_2, \ldots, a_n\}$ of objects, where each object $a_i$ is associated with two positive integers, namely a **value** $q(a_i)$ and a **weight** $w(a_i)$, $1 \leq i \leq n$; an integer $W$ (representing knapsack capacity).

Required: A subset $A' \subseteq A$ such that the total weight of the objects in $A'$ is at most $W$ and the total value of the objects in $A'$ is a maximum among all subsets satisfying the weight constraint.

**Lemma 4.2.** *The DMP-WC problem is an instance of the Knapsack problem.*

**Proof:** To see the correspondence between the DMP-WC problem and the Knapsack problem, we think of the tags as objects. The weight and profit of each tag correspond to weight and profit of each object in the Knapsack problem. The constraint on the total weight of the tags chosen corresponds to the constraint on the capacity of the Knapsack. The objective of the two problems is also the same (i.e., choosing a subset of items that maximize the total profit under the weight constraint). Thus, the DMP-WC problem can be solved as a Knapsack problem.

In general, the Knapsack problem is **NP**-hard. However, this computational intractability is due to large integers appearing as profits and weights (Garey & Johnson, 1979). It is known that the problem can be solved in pseudo-polynomial time using dynamic programming; that is, the running time of the dynamic programming algorithm is a polynomial function of the number of objects ($n$) and the value of the largest integer appearing in the input (Garey & Johnson, 1979). In the case of the DMP-WC problem, the largest possible weight is the number of normal instances and the largest possible profit is the number of outlier instances. Since these values are bounded by the size of the DMP-WC problem, any pseudo-polynomial time dynamic programming algorithm for the problem (Cormen et al., 2009) is indeed a *polynomial time*

algorithm for the problem. For the sake of completeness and since this is one of the algorithms that we studied experimentally, we provide the details of the dynamic programming algorithm for DMP-WC below.

---

**Algorithm 2:** Dynamic Programming Algorithm for DMP-WC.

> **Input :** Tags $\mathbb{U} = \{u_1, u_2, \ldots, u_n\}$; for each tag $u_i$, its profit $p(u_i)$ and weight $w(u_i)$; budget $W$ on total weight.
> **Output :** A subset $Y^*$ of $\mathbb{U}$ with the maximum profit under the constraint that the total weight of the tags in $Y^*$ is at most $W$.

**1** Let $P$ be an $(n+1) \times (W+1)$ array. (See the text for the interpretation of the values stored in the $P$ array. (The row and column indices of $P$ vary from 0 to $n$ and 0 to $W$ respectively.)

// Compute the optimal profit value.

**2** **for** $j = 0, 1, \ldots, W$ **do**
**3** $\quad$ $P[0, j] = 0$
**4** **end**
**5** **for** $i = 1, 2, \ldots, n$ **do**
**6** $\quad$ **for** $j = 0, 1, \ldots, W$ **do**
**7** $\quad\quad$ $P[i, j] = \max\{P[i-1, j], \ P[i-1, j - w(p_i)] + p(u_i)\}$
**8** $\quad$ **end**
**9** **end**

// Optimal profit is $P[n, W]$. Find an optimal solution $Y^*$.

**10** Let $Y^* = \emptyset$, temp $= P[n, W]$, $i = n$ and $j = W$
**11** **while** $i \geq 1$ **do**
**12** $\quad$ **if** temp $> P[i-1, j]$ **then**
**13** $\quad\quad$ Add $u_i$ to $Y^*$
**14** $\quad\quad$ temp $= P[i-1, j - w(u_i)]$
**15** $\quad\quad$ $j = j - w(u_i)$
**16** $\quad$ **end**
**17** $\quad$ $i = i - 1$
**18** **end**
**19** Return $P[n, W]$ (optimal profit) and $Y^*$ (optimal solution).

---

**A dynamic programming algorithm for DMP-WC:** Let $\mathbb{U} = \{u_1, u_2, \ldots, u_n\}$ denote the set of $n$ tags. Recall that for each tag $u_i$, its profit and weight are given by $p(u_i)$ and $w(u_i)$ respectively. Let $W$ denote the maximum total weight of the tags chosen. Without loss of generality, we assume that for each tag $u_i$, the weight of $u_i$, that is, $w(u_i)$ is at most $W$. (Tags whose weights are larger than $W$ cannot be used in the solution.) The dynamic programming algorithm uses a two dimensional table $P$ with $n + 1$ rows and $W + 1$ columns. For each $i$ and $j$, where $0 \leq i \leq n$ and $0 \leq j \leq W$, the entry $P[i, j]$ stores the maximum profit that can be realized using the subset of tags $\{u_1, u_2, \ldots, u_i\}$ under the constraint that the total weight of the chosen tags is at most $j$. Now, the equations to compute the entries of this table are as follows.

$$P[0, j] = 0, \quad 0 \leq j \leq W.$$
$$P[i, j] = \max\{P[i-1, j], P[i, j - w(u_i)] + p(u_i)\}, \quad 0 \leq j \leq W.$$

Once all the entries of the $P$ matrix are available, the optimal solution value is given by $P[n, W]$.

In the above computation, we employ the common convention that if the second index of the $P$ matrix is less than 0 (i.e., it represents an infeasible weight), the corresponding profit value is 0. The above equations can be used to compute the optimal profit for the DMP-WC problem. Using the values in the $P$ matrix, an optimal subset $\mathbb{U}^*$ of tags can be found. Pseudocode for the dynamic programming method is shown as Algorithm 2.

**Running time analysis:** To estimate the running time of Algorithm 2, we first note that the number of entries in the $P$ matrix is $O(nW)$, where $n = |\mathbb{U}|$ is the number of tags and $W$ is the weight budget. The running time of the algorithm is dominated by the time needed to compute all the entries of the $P$ matrix. From Step 7 of the algorithm, it is seen that each entry of the $P$ matrix can be computed in $O(1)$ time. So, the total time to compute all the entries and hence to obtain the optimal solution value is $O(nW)$. Note that the weight of any tag $u_i$ is at most $|\mathbb{N}|$, the number of normal instances. Thus, the value of $W$ is at most $n|\mathbb{N}|$. Hence, the overall running time of Algorithm 2 is $O(n^2|\mathbb{N}|)$, which is polynomial in the size of the problem. The following theorem summarizes the above discussion.

**Theorem 4.2.** *An optimal solution to the DMP-WC problem can be found in $O(n^2|\mathbb{N}|)$ time, where $n$ and $|\mathbb{N}|$ are respectively the number of tags and the number of normal instances.*

### 4.3 A Complexity Result for Multiple Descriptions

In this section, we show that the problem of obtaining multiple descriptions under weight constraints (i.e., the problem MDMP-WC defined in Section 3.2) is **NP**-complete. Since the notion of **NP**-completeness is for decision problems, we will assume that the problem specification also includes an additional integer parameter $\lambda$. The goal of the decision version of MDMP-WC is to determine whether there are $\sigma$ subsets $Y_1$, $Y_2$, ..., $Y_\sigma$ of tags satisfying the constraints of the MDMP-WC problem and the additional condition that the total profit of the descriptions is at least $\lambda$.

**A note about the reduction:** As mentioned earlier, the classical Knapsack problem is computationally intractable because the numbers specified as part of the problem (i.e., the profit and weights of given items) can be exponentially large in the number of given items (Garey & Johnson, 1979). Our efficient algorithm for the DMP-WC problem, which requires us to find one subset of tags, was obtained by observing that the problem reduces to a special case of the Knapsack problem where the weight (and profit) values are numbers whose value is bounded by a polynomial function of the size of the problem instance (Section 4.2). As we show below, in the case of the MDMP-WC problem, the computational intractability arises because of the need for multiple knapsacks. For this reason, it is difficult to use a direct reduction from the classical (single) Knapsack problem to prove the **NP**-hardness of the MDMP-WC problem. Instead, it is more convenient to use a reduction from a known **NP**-hard problem which involves partitioning a set of integers into many subsets (which represent multiple knapsacks). One such problem is 3-PARTITION (Garey & Johnson, 1979) defined below.

**3-Partition:**

Given: Positive integers $m$ and $B$, where $B$ is bounded by a polynomial function of $m$, a set $A = \{z_1, z_2, \ldots, z_{3m}\}$ of $3m$ positive integers such that $\sum_{i=1}^{3m} z_i = mB$ and for $1 \le i \le 3m$, $B/4 < z_i < B/2$.

Question: Can $A$ be partitioned into $m$ subsets $A_1$, $A_2$, ..., $A_m$ such that for each $A_j$, the sum of the integers in $A_j$ is exactly $B$?

It is shown in (Garey & Johnson, 1979) that 3-PARTITION is **NP**-complete. From the constraints on the values of the integers in $A$, note that whenever there is a solution to a 3-PARTITION instance, each subset $A_j$ must contain *exactly three* integers.

**Theorem 4.3.** *The MDMP-WC problem is **NP**-complete.*

**Proof:** It can be seen that MDMP-WC is in **NP** since given a solution, one can efficiently verify that the subsets are pairwise disjoint, each subset satisfies the weight constraint and the total profit from all the subsets is at least $\lambda$. We prove the **NP**-hardness of MDMP-WC through a reduction from the 3-PARTITION problem. Given an instance $I$ of the 3-PARTITION problem, we construct an instance $I'$ of MDMP-WC problem as follows. As before, we use $G(V_{\mathbb{N}+\mathbb{O}}, V_{\mathbb{T}}, E)$ to denote the bipartite graph of the MDMP-WC problem.

1. The universe of tags $\mathbb{U} = \{u_1, u_2, \ldots, u_{3m}\}$ has $3m$ elements, with tag $u_i$ corresponding to integer $z_i$ in $A$, $1 \le i \le 3m$. Thus, the node set $V_{\mathbb{T}}$ of $G$ is of size $3m$.
2. With each tag $u_i$, we associate a unique set of $z_i$ normal instances, denoted by $N_i^j$, $1 \le j \le a_i$, and one outlier instance $O_i$, $1 \le i \le 3m$. (Thus, each normal and outlier instance is associated with exactly one

tag.) This construction ensures that the weight of tag $u_i$ is $z_i$ and its profit is 1, for $1 \leq i \leq 3m$. The total numbers of normal and outlier instances created are $mB$ and $3m$ respectively. In other words, the node set $V_{\mathbb{N}+\mathbb{O}}$ of $G$ has a total of $m(B+3)$ nodes, with $mB$ nodes in $V_{\mathbb{N}}$ and $3m$ nodes in $V_{\mathbb{O}}$.

3. The edge set $E$ of $G$ is constructed as follows. For each tag $u_i$, there is an edge between $u_i$ and its associated normal instances and outlier instance, $1 \leq i \leq 3m$. (Thus, the degree of each normal and outlier instance in the underlying bipartite graph is exactly 1.)

4. The number $\sigma$ of subsets of tags required is set to $m$, the budget on the weight of each required subset is set to $B$ and the required profit is set to $3m$.

This completes the construction of the instance $I'$ of the MDMP-WC problem. Since the value of $B$ is a polynomial function of $m$, the construction can be done in polynomial time. We will now show that there is a solution to the instance $I'$ of MDMP-WC iff there is a solution to the 3-PARTITION instance $I$.

**Part 1:** Suppose there is a solution to the 3-PARTITION instance $I$ consisting of sets $A_1$, $A_2$, ..., $A_m$. Recall that in this solution, each set $A_j$ has exactly three integers. We construct a solution to the MDMP-WC instance $I'$ as follows. For each set $A_j$, we construct a subset $Y_j$ consisting of three tags, in the following manner: if $A_j$ has the integers $z_a$, $z_b$ and $z_c$, the set $Y_j$ has the tags $u_a$, $u_b$ and $u_c$. Recall that the weight of each tag $u_i$ is $z_i$ and the profit of each tag is 1. Since the sum of the integers in $A_j$ is $B$, the total weight of each subset $Y_j$ is $B$. Since there are three tags in $Y_j$ and the profit of each tag is 1, the profit of $Y_j$ is 3 and the total profit over the $m$ subsets of tags is $3m$. We thus have a solution to the MDMP-WC instance $I'$.

**Part 2:** Suppose there is a solution to the MDMP-WC instance $I'$. We have the following claim.

Claim 1: In the solution to the MDMP-WC instance, each set $Y_j$ has exactly three tags and their total weight is equal to $B$.

Proof of Claim 1: The total weight of all the tags is $mB$ and this is weight is partitioned over the $m$ pairwise disjoint subsets $Y_1$, $Y_2$, ..., $Y_m$. By the weight constraint, the weight of each tag set must be at most $B$. If a set $Y_j$ has a weight less than $B$, then some other set $Y_r$ must have a weight exceeding $B$, thus violating the weight constraint. Thus, each set $Y_j$ must have a total weight of exactly $B$. Also, by the specification of the 3-PARTITION problem, each tag has a weight which is *greater than $B/4$* but *less than $B/2$*. Therefore, for the weight of a set $Y_j$ to be exactly $B$, the number of tags in $Y_j$ must be exactly three. This completes a proof of Claim 1.

In view of Claim 1, we can construct a solution to the 3-PARTITION instance $I$ as follows. For each set $Y_j$, construct a subset $A_j$ of $A$ as follows. If $Y_j$ has the tags $u_a$, $u_b$ and $u_c$, let $A_j$ contain the integers $z_a$, $z_b$ and $z_c$. By Claim 1, the sum of the integers in $A_j$ is exactly $B$. Moreover, since the subsets in the solution to the MDMP-WC instance are pairwise disjoint, the subsets $A_1$, $A_2$, ..., $A_m$ are also pairwise disjoint. Thus, we have a solution to the 3-PARTITION instance $I$, and this completes our proof of Theorem 4.3.

## 5 Experiments

From our experiments, we aim to answer the following questions:

- What explanations do our three formulations find for a variety of different data modalities (image, text and database)? We address this by comparing our explanations in domains where a strong ground truth is known.
- Can we use our method for novel scenarios such as finding contrastive explanations to explain the differences between the predictions of multiple outlier detection algorithms?

We use three data sets, with one for each modality: CelebA (Liu et al., 2015) (Images), COMPAS (Angwin et al., 2016) (Databases) and HateXPlain (Mathew et al., 2021)) (Text). We next discuss these datasets in terms of content, how anomalies are found and the ground truths we will compare our methods against. As will be pointed out in the related work section (i.e., Section 6), existing work for XAD does not simultaneously use semantic tags, generate contrastive explanations and provide a comparable global level explanation; hence, there are no direct comparisons to be made. The closest work to our own is on contrastive pattern

mining (Chen et al., 2022) which returns poor results for explanation as there is no coverage requirement. See Section 6 for a discussion on contrastive pattern mining.

### 5.1 Descriptions of Data Sets

This section can be skipped by readers familiar with these data sets.

**CelebA (Liu et al., 2015)** This data set consists of 202,599 images of celebrities, each of which can have up to 40 tags given by a human annotator. We add an additional 14 tags generated by the Deepface (Serengil & Ozpinar, 2021) facial analysis framework. An example of typical normal and outlier points found is shown in Figure 3. We generate outliers with this data using both Deep SVDD (Ruff et al., 2018) and a deep convolutional auto encoder (DCAE) using a well known deep learning architecture (Subramanian, 2020). In previous work on fairness using Deep SVDD (Zhang & Davidson, 2021), it was found by visualization of the most and least anomalous instances that the definition of normal celebrities were white females and the outliers were overwhelmingly people of color and male. Our work will generate explanations that contrast outliers and normal images (see Tables 2, 3 and 4). We also explain the differences between the normal predictions that SVDD and CVAE outlier detection algorithms make (see Table 6).

**COMPAS (Angwin et al., 2016)** is a classification system that scores a criminal defendant's chance of recidivism by a decile score. The higher the recidivism score, the greater is the belief that the person will reoffend after being released from prison. To this score is added whether the individual did in fact reoffend within a two year period (Angwin et al., 2016). We consider as being unfairly and harshly treated (outliers) individuals who have a decile score of 10 but who did not reoffend within two years. These are effectively false positives (predicted to reoffend but did not). We generate a contrastive explanation to the group of individuals with a decile score of 1 but who did reoffend within two years (too leniently treated) which are effectively false negatives. We compare our explanations against the findings by ProPublica (Angwin et al., 2016).

**HateXPlain (Mathew et al., 2021)** is a corpus of 20,148 Twitter and Gab posts. HateXPlain classifies text as "normal speech", "hate speech", or "offensive speech" for specific categories (race, sex, etc.) (Devlin et al., 2018). We take the subset of "normal speech" for the "African" category as normal text, and the subset of "hate speech" for the "African" category as outliers.

### 5.2 CelebA - Explaining Deep SVDD Predictions

Here we identify outliers in the images of celebrities and then explain the output of the AD algorithm using the tags for each image. We focus on the results produced using the Deep SVDD (Ruff et al., 2018) AD algorithm. Examples of the normal and outlier points found by this algorithm are shown in Figure 3.

We applied our first formulation to the AD algorithm's output and generated the explanations in Table 2. We report the fraction of instances covered in each collection/group (center columns) and the fraction of the entire population that have each tag (right column). These explanations match our visual understanding of the outliers and the understanding of others (Zhang & Davidson, 2021). Several tags in the explanation are highly contrastive such as `Race:Black`, `Black_hair`, `Eyeglasses` and `Wearing_hat` for the outlier group and `Blond_Hair` and `Brown_Hair` for the normal group. However, some tags in the explanation are not particularly contrastive, such as `Male` for the outlier group that occurs in 32.5% of the entire population and 40% of the outlier group and 25% in the normal group. Similarly, the tag `Race:White` in the normal explanation occurs in 55.5% of the population but 75% of the normal group and 36% of the outlier group.

Our second formulation aims to overcome this limitation to find a strongly contrastive explanation which is shown in Table 3. Interestingly, we see that the not strongly contrastive tag `Male` found in Formulation #1 has been replaced by a more discriminating surrogate (`Mustache`) for the outlier group, and the non-contrastive tag `Race:White` for the normal group found in Formulation #1 is replaced by tags `Blond_Hair`, `Rosy Cheeks` which are typically found in people with fair complexions.

## Most Normal Smiling Instances

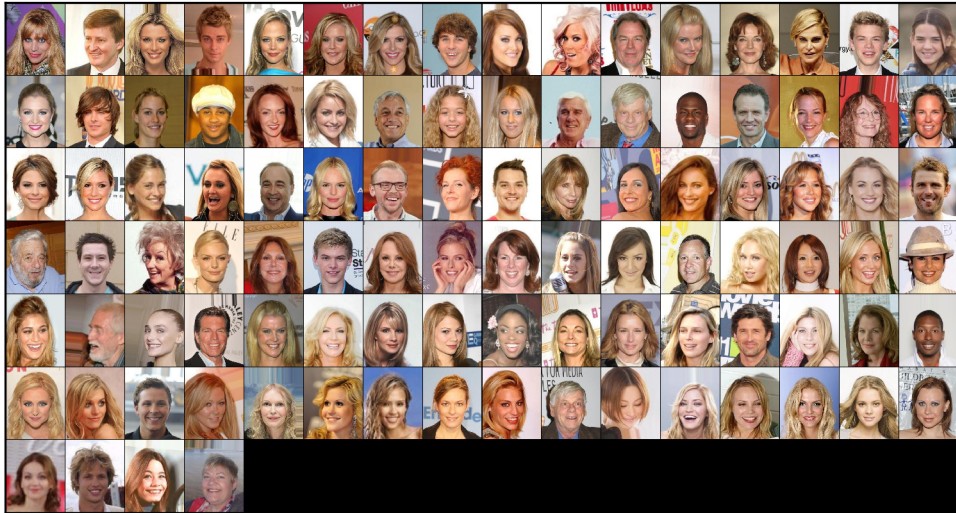

## Most Anomalous Smiling Instances

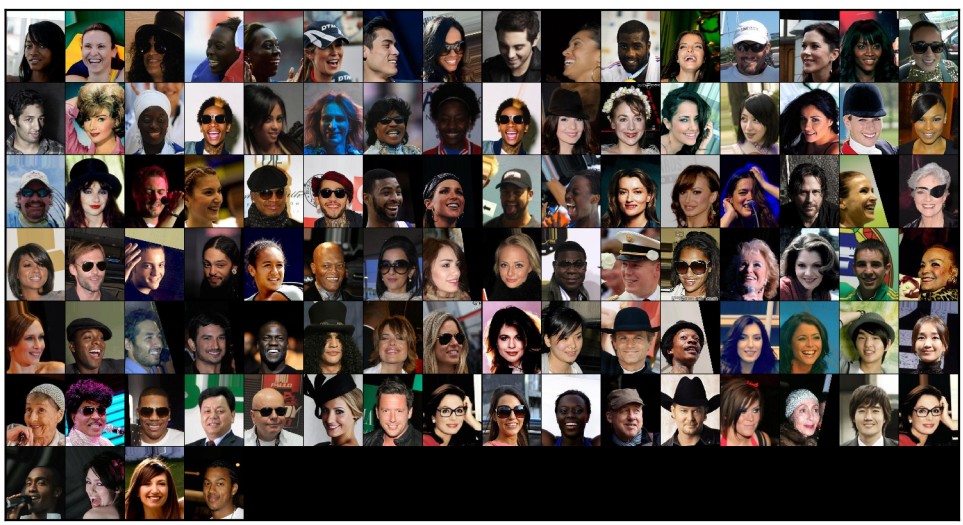

Figure 3: Top 100 most normal and most anomalous instances found by Deep SVDD based solely on image data. Note that although the normal and outlier points backgrounds are different, there are significant differences in the faces of the two types of people.

Our third formulation allows finding multiple explanations if they exist. The number of explanations found is pre-set as a hyperparameter. We see there are three dominant explanations involving (i) `Black hair`, (ii) `Wearing Hat`, and (iii) `Eyeglasses` as shown in Table 4.

### 5.3 CelebA - Explaining Differences Between Deep SVDD and DCAE

Here we test the use of our method for finding the contrasts/differences between the predictions of Deep SVDD and a deep convolutional autoencoder (DCAE) on the same data set. We focus on the challenging but important problem of explaining the definitions of normality between the algorithms. The top part of Figure 3 shows Deep SVDD's typical normal images and Figure 4 shows some normal images for DCAE. We find in Table 5 interestingly that there is a strong difference between the definitions of normality for both

## AE Most Normal Smiling Instances

Figure 4: Top 100 normal images found using a DCAE. c.f. Figure 3 top.

Table 2: Formulation 1. The explanations of CelebA instances outliers/normal points found using Deep SVDD. Compare with Table 3. For each collection (outlier/normal/population) we state the fraction of that collection having the tag.

| Outlier Explanation | Instance Coverage Outlier Group \| Normal Group | Population Coverage |
|---|---|---|
| Black_Hair | 36.0% \| 2.0% | 19.0% |
| Eyeglsses | 22.0% \| 2.0% | 12.0% |
| Male | 40.0% \| 25.0% | 32.5% |
| Wearing_Hat | 19.0% \| 2.0% | 10.5% |
| Race: Black | 18.0% \| 3.0% | 10.5% |

| Normal Explanation | Instance Coverage Normal Group \| Outlier Group | Population Coverage |
|---|---|---|
| Blond_Hair | 44.0% \| 2.0% | 23.0% |
| Race: White | 75.0% \| 36.0% | 55.5% |
| Arched_Eyebrows | 46.0% \| 16.0% | 31.0% |
| Gender: Woman | 68.0% \| 35.0% | 51.5% |
| Brown_Hair | 26.0% \| 4.0% | 15.0% |

algorithms. The deep SVDD algorithm associates normality with blond individuals and the DCAE algorithm with black haired individuals. However, the remaining tags are not particularly discriminatory. Our second formulation's results (see Table 6) ensure strong contrastive results and we see that the DCAE and Deep SVDD algorithms are quite different in terms of race, gender and hair color. It seems that the AE defines normality as men where as the SVDD algorithm defines normality centered around being a woman.

### 5.4 COMPAS: False Positives against False Negatives

Here we focus on explaining what characterizes individuals who are predicted to reoffend but did not (false positives who are unfairly treated) compared to those who were not predicted to reoffend but did (false negative who are favorably treated). For explanation we use tags for age, sex, race, crime severity (felony

Table 3: Formulation 2. The explanations of CelebA instances outliers/normal points found using Deep SVDD. Compare with Table 2. For each collection (outlier/normal/population) we state the fraction of that collection having the tag.

| Outlier Explanation | Instance Coverage Outlier Group \| Normal Group | Population Coverage |
|---|---|---|
| Black_Hair | 36.0% \| 2.0% | 19.0% |
| Eyeglsses | 22.0% \| 2.0% | 12.0% |
| Wearing_Hat | 19.0% \| 2.0% | 10.5% |
| Race: Black | 18.0% \| 3.0% | 10.5% |
| Mustache | 10.0% \| 1.0% | 5.5% |

| Normal Explanation | Instance Coverage Normal Group \| Outlier Group | Population Coverage |
|---|---|---|
| Blond_Hair | 44.0% \| 2.0% | 23.0% |
| Brown_Hair | 26.0% \| 4.0% | 15.0% |
| Rosy_Cheeks | 20.0% \| 2.0% | 11.0% |
| Gray_Hair | 8.0% \| 2.0% | 5.0% |
| Pale_Skin | 2.0% \| 0.0% | 1.0% |

Table 4: Formulation 3. Multiple explanations of CelebA instances outliers/normal points found using Deep SVDD. Compare with Table 3.For each collection (outlier/normal/population) we state the fraction of that collection having the tag.

| Outlier Explanations | Instance Coverage Outlier Group \| Normal Group | Population Coverage |
|---|---|---|
| Group #1 | | |
| Black_Hair | 36.0% \| 2.0% | 19.0% |
| Bushy_Eyebrows | 10.0% \| 5.0% | 7.5% |
| Blurry | 8.0% \| 3.0% | 5.5% |
| Race: Indian | 2.0% \| 0.0% | 1.0% |
| Emotion: Surprise | 1.0% \| 0.0% | 0.5% |
| Group #2 | | |
| Wearing_Hat | 19.0% \| 2.0% | 10.5% |
| 5_o_Clock_Shadow | 9.0% \| 5.0% | 7.0% |
| Goatee | 8.0% \| 1.0% | 4.5% |
| Emotion: Angry | 3.0% \| 2.0% | 2.5% |
| Group #3 | | |
| Eyeglasses | 22.0% \| 2.0% | 12.0% |
| Race: Black | 18.0% \| 3.0% | 10.5% |
| Mustache | 10.0% \| 1.0% | 5.5% |
| Emotion: Sad | 5.0% \| 1.0% | 3.0% |
| Sideburns | 4.0% \| 1.0% | 2.5% |

or misdemeanor), and the specific crime committed. We note that the crime labels are taken directly from the COMPAS dataset and often use many abbreviations. Note our analysis is different from the ProPublica analysis (Angwin et al., 2016) which only focuses on the false positive group and statistics for it. ProPublica found strong racial disparity in COMPAS predictions between white and black defendants in this group.

Using Formulation 1 (see Table 7), we observe results consistent with ProPublica findings. Our explanation shows that defendants who are treated too harshly by COMPAS are largely young African-Americans

Table 5: Formulation 1. The normal group explanations of Deep SVDD versus DCAE. For each collection (AE-Normal/SVDD-Normal/population) we state the fraction of that collection having the tag.

| AE Explanation | Instance Coverage AE Group\|SVDD Group | Population Coverage |
|---|---|---|
| Black_Hair | 32.0% \| 2.0% | 17.0% |
| Young | 90.0% \| 72.0% | 81.0% |
| Heavy_Makeup | 62.0% \| 46.0% | 54.0% |
| Wearing_Lipstick | 74.0% \| 63.0% | 68.5% |
| Attractive | 66.0% \| 57.0% | 61.5% |

| SVDD Explanation | Instance Coverage SVDD Group\|AE Group | Population Coverage |
|---|---|---|
| Blond_Hair | 43.0% \| 1.0% | 22.0% |
| Race: White | 78.0% \| 40.0% | 59.0% |
| Emotion: Happy | 75.0% \| 55.0% | 65.0% |
| Mouth_Slightly_Open | 73.0% \| 55.0% | 64.0% |
| Bags_Under_Eyes | 22.0% \| 11.0% | 16.5% |

Table 6: Formulation 2. The normal group explanations of Deep SVDD versus DCAE. For each collection (AE-Normal/SVDD-Normal/population) we state the fraction of that collection having the tag.

| AE Explanation | Instance Coverage AE Group\|SVDD Group | Population Coverage |
|---|---|---|
| Black_Hair | 32.0% \| 2.0% | 17.0% |
| 5_o_Clock_Shadow | 6.0% \| 4.0% | 5.0% |
| Pale_Skin | 5.0% \| 1.0% | 3.0% |
| Sideburns | 2.0% \| 1.0% | 1.5% |

| SVDD Explanation | Instance Coverage SVDD Group\|AE Group | Population Coverage |
|---|---|---|
| Blond_Hair | 43.0% \| 1.0% | 22.0% |
| Chubby | 8.0% \| 2.0% | 5.0% |
| Gray_Hair | 7.0% \| 0.0% | 3.5% |
| Emotion: Fear | 7.0% \| 4.0% | 5.5% |
| Double_Chin | 6.0% \| 0.0% | 3.0% |

who were charged with felonies. Further, consistent with ProPublica findings, our explanation shows that defendants who are treated too leniently by COMPAS are largely older Caucasian defendants charged with misdemeanors.

It is with Formulation 2 (see Table 8) and Formulation 3 (see Table 9) that we show insights not previously found. We observe that the results of Formulation 2 emphasize how defendants treated too harshly tended to be young and having been charged for more drug offenses, where as defendants treated too leniently are older and were found to have been charged with driving offenses. There is a strong correlation between race and the types of charges, indicating a selection policing bias. Using Formulation 3, we find explanations for defendants treated too harshly, and set the number of explanations $k = 3$ to be consistent with our configuration with CelebA. We see that the three dominant explanations for COMPAS for Formulation 3 are: Less than 25, Possession of Cocaine, and people arrested but no charges filed.

Table 7: Formulation 1. The explanation of COMPAS False Positives (Harshly Treated) versus False Negatives (Leniently Treated). For each collection (False-Positive/False-Negative/Population) we state the fraction of that collection having the tag.

| False Positive Explanations | Instance Coverage False Positive Group \| False Negative Group | Population Coverage |
|---|---|---|
| African-American | 68.2% \| 29.3% | 37.8% |
| Less than 25 | 23.5% \| 1.0% | 5.9% |
| Charge Degree: Felony | 71.8% \| 49.8% | 54.6% |
| Age: 25–45 | 71.8% \| 51.5% | 55.9% |
| "arrest case no charge" | 25.9% \| 9.4% | 13.0% |

| False Negative Explanations | Instance Coverage False Negative Group \| False Positive Group | Population Coverage |
|---|---|---|
| Age: Greater than 45 | 47.6% \| 4.7% | 38.3% |
| Caucasian | 46.3% \| 22.4% | 41.4% |
| Charge Degree: Misdemeanor | 50.2% \| 28.2% | 45.4% |
| Battery | 25.7% \| 9.4% | 22.2% |
| Hispanic | 15.6% \| 7.1% | 13.8% |

Table 8: Formulation 2. The explanation of COMPAS False Positives versus False Negatives. For each collection (False-Positive/False-Negative/Population) we state the fraction of that collection having the tag.

| False Positive Explanations | Instance Coverage False Positive Group \| False Negative Group | Population Coverage |
|---|---|---|
| Less than 25 | 23.5% \| 1.0% | 5.9% |
| Grand Theft (Motor Vehicle) | 3.5% \| 1.6% | 2.0% |
| "Pos Cannabis W/Intent Sel/Del" | 2.4% \| 0.7% | 1.0% |
| Possession of Alprazolam | 2.4% \| 1.0% | 1.3% |
| "Burglary Conveyance Unoccup" | 2.4% \| 1.0% | 1.3% |

| False Negative Explanations | Instance Coverage False Negative Group \| False Positive Group | Population Coverage |
|---|---|---|
| Age: Greater than 45 | 47.6% \| 4.7% | 38.3% |
| "Other" | 8.5% \| 2.4% | 7.1% |
| Driving Under The Influence | 5.9% \| 0.0% | 4.6% |
| "Felony Driving While Lic. Suspd" | 3.3% \| 0.0% | 2.6% |
| "Viol Injunct Domestic Violence" | 2.0% \| 0.0% | 1.5% |

## 5.5 HateXPlain

The reader is warned that the content of this data set is highly offensive but is a classic text explanation data set. Here we generate explanations for the words that are most hateful in the HateXPlain dataset. In Formulation 1 (See Table 10) we observe many derogatory and hateful words being contrasted against more individually neutral words. Notably, the n-word ending in an 'r' is the strongest form of hate speech found by the explanation. However, the n-word ending in an 'a' is the strongest explanation for texts that are classified as normal speech. This supports generalizations by Mathew et al. (2021) regarding how many texts in the dataset reflect the n-word being used within the African-American community as a pronoun. Results for Formulations 2 and 3 are shown in Tables 11 and 12 respectively.

Table 9: Formulation 3. Multiple explanations of COMPAS false positives when contrasted against false negatives. For each collection (False-Positive/False-Negative/Population) we state the fraction of that collection having the tag.

| False Positive Explanations | Instance Coverage False Positive Group \| False Negative Group | Population Coverage |
|---|---|---|
| | Group #1 | |
| Less than 25 | 23.5% \| 1.0% | 5.9% |
| Grand Theft in the 3rd Degree | 9.4% \| 5.2% | 6.1% |
| "Felony Petit Theft" | 2.4% \| 0.0% | 0.5% |
| Felony Battery w/Prior Convict. | 1.2% \| 0.0% | 0.3% |
| Possession of Cannabis $50 | 1.2% \| 0.3% | 0.5% |
| | Group #2 | |
| Possession of Cocaine | 7.1% \| 4.6% | 5.1% |
| Grand Theft (Motor Vehicle) | 3.5% \| 1.6% | 2.0% |
| "Pos Cannabis W/Intent Sel/Del" | 2.4% \| 0.7% | 1.0% |
| Possession of Alprazolam | 2.4% \| 1.0% | 1.3% |
| "Burglary Conveyance Unoccup" | 2.4% \| 1.0% | 1.3% |
| | Group #3 | |
| "arrest case no charge" | 25.9% \| 9.4% | 13.0% |
| "Escape" | 1.2% \| 0.3% | 0.5% |

Table 10: Formulation 1. Explanations of HateXPlain hate speech and normal speech. For each collection (Hate-Speech/Normal-Speech/Population) we state the fraction of that collection having the tag.

| Hate Speech Explanation | Instance Coverage Hate Speech Group \| Normal Speech Group | Population Coverage |
|---|---|---|
| n*gger | 76.2% \| 24.4% | 72.8% |
| f*ck | 13.7% \| 6.8% | 13.3% |
| jew | 8.1% \| 1.8% | 7.7% |
| k*ke | 5.7% \| 0.0% | 5.3% |
| f*ggot | 5.1% \| 0.5% | 4.8% |

| Normal Speech Explanation | Instance Coverage Normal Speech Group \| Hate Speech Group | Population Coverage |
|---|---|---|
| n*gga | 29.9% \| 4.6% | 6.2% |
| black | 38.0% \| 16.8% | 18.2% |
| <user> | 31.7% \| 13.1% | 14.3% |
| white | 37.6% \| 27.9% | 28.5% |
| people | 16.3% \| 7.5% | 8.0% |
| person | 6.8% \| 1.3% | 1.7% |

Table 11: Formulation 2. Explanations of HateXPlain hate speech and normal speech. For each collection (Hate-Speech/Normal-Speech/Population) we state the fraction of that collection having the tag.

| Hate Speech Explanation | Instance Coverage
Hate Speech Group \| Normal Speech Group | Population Coverage |
|---|---|---|
| jew | 8.1% \| 1.8% | 7.7% |
| k*ke | 5.7% \| 0.0% | 5.3% |
| f*ggot | 5.1% \| 0.5% | 4.8% |
| sand | 4.5% \| 0.0% | 4.2% |
| re*ard | 3.7% \| 0.0% | 3.4% |

| Normal Speech Explanation | Instance Coverage
Normal Speech Group \| Hate Speech Group | Population Coverage |
|---|---|---|
| yo | 2.7% \| 0.3% | 0.5% |
| <woozy face emoji> | 2.7% \| 0.1% | 0.2% |
| <leading face emoji> | 2.3% \| 0.0% | 0.1% |
| tf | 1.8% \| 0.0% | 0.1% |
| <time> | 1.4% \| 0.0% | 0.1% |

Table 12: Formulation 3. Multiple explanations of HateXPlain hate speech when contrasted against normal speech. For each collection (Hate-Speech/Normal-Speech/Population) we state the fraction of that collection having the tag.

| Hate Speech Explanation | Instance Coverage
Hate Speech Group \| Normal Speech Group | Population Coverage |
|---|---|---|
| Group #1 | | |
| k*ke | 5.7% \| 0.0% | 5.3% |
| sand | 4.5% \| 0.0% | 4.2% |
| re*ard | 3.7% \| 0.0% | 3.4% |
| back | 3.0% \| 1.4% | 2.9% |
| race | 2.9% \| 0.0% | 2.7% |
| Group #2 | | |
| f*ggot | 5.1% \| 0.5% | 4.8% |
| look | 4.0% \| 1.8% | 3.9% |
| good | 3.4% \| 1.4% | 3.2% |
| america | 2.9% \| 1.4% | 2.8% |
| little | 1.6% \| 0.5% | 1.5% |
| Group #3 | | |
| jew | 8.1% \| 1.8% | 7.7% |
| take | 3.5% \| 0.5% | 3.3% |
| rape | 3.2% \| 1.4% | 3.1% |
| anoth | 2.3% \| 0.9% | 2.2% |
| africa | 2.2% \| 0.9% | 2.1% |

# 6   Related XAD Work

We begin by overviewing the need for contrastive explanations and then move onto the areas of XAI and XAD that touch upon contrastive explanations.

**Contrastive Approaches in Human Explanation.** In an authoritative survey, Miller (2019) explores how humans explain things to each other and makes a core finding that contrast is a critical part of human explanation. This provides a solid motivation for our work as in the end the machine is explaining to a human. However, Miller (2019) also finds that very little work in XAI directly addresses contrastive explanations. To address this, his later work (Miller, 2021) explores a causal model but only for instance-level explanation in the classification setting.

There are several core areas where contrast is used in anomaly detection and XAI which we discuss below. Contrastive methods have been used to generate outliers and in XAI and XAD. Several authoritative surveys of the both these areas find that the area of model-level explainable anomaly detection is not well studied.

**Counterfactual Instance-Level XAD.** Counterfactual methods have been used to generate a contrastive normal instance for each outlier. For example in one such method (Sipple & Youssef, 2022), for each identified anomaly they use Integrated Gradients (IG) techniques to attribute an anomaly score to each feature and provide a contrastive nearest normal instance as explanation. However, these methods are instance level not model level explanations and are only suitable for settings where the input space is human interpretable as the outlier and the counterfactual are compared to generate the explanation. Further, generating realistic counterfactuals is a challenging problem. In comparison, our work is model level and uses semantic tags as a mechanism to generate explanations. As such, it relies on interpretable tags but not on the feature space being interpretable. Hence, our method can be applied to data modalities such as images, graphs and even text (see Table 1).

**Survey on XAD.** A recent survey on explainable anomaly detection (Li et al., 2024) overviews the field. Here we touch upon the parts of the field relevant to our work. This survey (Li et al., 2024) assesses XAD along six dimensions: (i) When explanation occurs, (ii) what level of granularity is the explanation applied to, (iii) model agnostic or model specific, (iv) feature or sample based, (v) computational technique used, and (vi) applicable to static or streaming data. Our work is an example of a post-hoc, global/model level and agnostic approach to explanation; it attempts to explain the entire output of any AD algorithm. Further, like most XAD techniques, it is applicable to static (not streaming) data. However, it is quite different from existing work for several reasons. Firstly, unlike existing methods that use the underlying features to explain the outliers, our work uses semantic tags (which may or may not be derived from the features). This means our work can be applied to the results of deep learning methods relatively easily so long as there are semantic tags for each instance. To our knowledge, no other XAD method takes this approach. Further, the computational technique we use to discover explanation is substantially different from existing work. Most of the known methods attempt to perform some underlying computation on the AD algorithm/function (such as perturbation to identify an explanation) where as our work formulates the problem as a computation on a bipartite graph that is similar to the knapsack problem. This allows us to use exact and efficient polynomial time algorithms based on dynamic programming. Finally, the ability to explain in a contrastive manner both anomalies and normal definitions has not been well studied.

**Other Computational Techniques That Could Be Used for Contrastive XAD.** Perhaps the most related work to our own is contrastive pattern mining (Chen et al., 2022) which attempts to find items frequent in one data set but rare in another. Though this work was not designed for explanation in AD it could be used for that purpose. However, that work has no coverage interpretations and experimentally produces poor results for explanation.

# 7   Conclusions

Anomaly and outlier detection is extensively used in AI and is often applied to policing and auditing in areas of substantial impact. However, XAD (Explainable Anomaly Detection) is a relatively understudied area given the importance of the topic. We propose the novel idea of finding a contrastive explanation, that is, a

set of tags that are common in say outliers but rare in normal points. By flipping the problem we can find explanations for the normal points as well. Using this idea, we develop three formulations.

We formulate explanation as a computation on a bipartite graph shown in Figure 1 with the aim of selecting a collection of tags that most explains the outlier points (by covering the edges incident on the outlier points) and minimally explains the normal points (by covering the edges incident on the inlier points). We provide three formulations with efficient algorithms for the first two and a proof of computational intractability for the third. Future work will explore designing efficient approximation algorithms for the third formulation. Our first and simplest formulation finds a contrastive explanation and we designed a simple linear time exact algorithm that finds an optimal explanation in that it selects the most tags incident on outlier points less tags incident on normal points. However, without any strong requirements, though this explanation scheme was useful we found that some tags were superfluous for explanation. Our second formulation addresses this concern by introducing the notion of a strongly contrastive explanation which tightly upper bounds how many of the tags used for explanation for say outliers can explain normal points. This creates a much more useful explanation but yields a more challenging computation. We formulate this version as a form of knapsack and create an exact and polynomial-time dynamic programming algorithm. Our two previous formulations find just a single explanation; as a novel variation, we create a third formulation that builds upon the second by creating $k$ different explanations. Unfortunately, this problem is computationally intractable; however, it can be formulated as an ILP and solved using large-scale solvers such as Gurobi.

To demonstrate the versatility of our approaches, we explored their application on a variety of data modalities: images (Celebrity A dataset), databases (COMPAS dataset) and text (HateXplain data set). For all three datasets, known ground truth explanations are well established and we ascertained that our methods are suitable for reproducing them and finding more nuanced explanations not previously published. Our methods successfully explained that outliers found using Deep SVDD in the Celebrity A dataset were overwhelmingly males and people of color as others have found in the work in fairness (Zhang & Davidson, 2021). Our third formulation extended these insights by showing that there exists a number of distinct types of outliers. A ProPublica article (Angwin et al., 2016) found that the AI tool Compas which predicted inmates' chances of reoffending was biased against minorities. Our first formulation discovered this insight and our strongly contrastive explanations found even more nuanced differences in terms of the crime. For the HateXPlain data set, we found that racial slurs are as expected used extensively for hate speech but that even regular speech uses slurs but in a different manner. Since no XAD algorithms to our knowledge use semantic tags as a basis of explanation, we do not compare against any known baselines.

**Acknowledgments.** We thank the TMLR referees and the Action Editor for providing very helpful feedback. This work was supported in part by NSF Grants IIS-1908530 and IIS-1910306 entitled "Explaining Unsupervised Learning: Combinatorial Optimization Formulations, Methods and Applications" and IIS-2310481 titled "Towards Fair Outlier Detection". We thank Google for their gift entitled, "Towards Fair and Explainable Learning".

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
