# OpenReview forum: "CXAD: Contrastive Explanations for Anomaly Detection: Algorithms, Complexity Results and Experiments"
_TMLR — Accepted by TMLR_

### Review · Reviewer_GHZM · 2025-03-16

**Summary Of Contributions:**

The paper addresses the problem of explainable anomaly detection (XAD) and highlights limitations of existing explainable AI (XAI) techniques in this context. The authors propose a contrastive explanation framework for anomaly detection, which aims to differentiate outliers from normal instances using interpretable semantic tags. Three formulations are introduced: (1) Contrastive Explanation – a model-agnostic explanation maximizing contrastive tags; (2) Strongly Contrastive Explanation – a constrained version ensuring stronger separation between normal and anomalous groups; and (3) Multiple Strongly Contrastive Explanations – A multi-explanation formulation that handles datasets with multiple anomaly types. Theoretical complexity analysis is provided, demonstrating that two of the formulations are solvable in polynomial time, while the third is NP-hard. The paper also presents experimental results on datasets across different modalities (images, text, and databases) to validate the approach.

**Audience:**

Yes

**Broader Impact Concerns:**

None.

**Claims And Evidence:**

Yes

**Requested Changes:**

I suggest that the authors clarify their assumptions regarding the tags and discuss potential practical methods for generating these tags.

Additionally, please comment on possible future work addressing the varying degrees of contribution from different tags.

**Strengths And Weaknesses:**

Strength:

The paper introduces a new contrastive approach specifically designed for anomaly detection, an area where explainability is crucial but underexplored.

Theoretical complexity analysis is rigorous, demonstrating that two formulations are solvable efficiently, while the third is NP-hard. The use of bipartite graphs and knapsack-style optimization problems provides a strong computational foundation.

Weakness:

The proposed definition of contrast explanations is interesting; however, additional clarification and justification would be helpful. Importantly, similar to traditional XAI techniques, the tags need to be physically meaningful and clearly indicative of the anomaly/outlier predictions. One may question the practicality of generating these tags, especially when dealing with high-dimensional data. For example, the paper uses MNIST for illustration, treating pixels as tags. However, pixels may not be good options for tags, since even minor pixel shifts could challenge the feasibility of the proposed approach.

Correspondingly, a significant concern arises about what the proposed method actually explains. For example, in the experiments, the proposed method seems less focused on directly explaining the behavior of the Deep SVDD algorithm and more centered on highlighting a potential relationship between Deep SVDD and the tags. While exploring such a relationship can indeed offer insights into Deep SVDD's behavior, it relies heavily on the assumption that the tags are strongly correlated with the predictive input features. The authors should explicitly state this assumption and clarify that their method depends on predefined, high-quality tags closely associated with these predictive features.

Another point of concern is the current formulation, which treats all edges equally. This uniform approach could be limiting since it overlooks the varying degrees of contribution different features or tags might have. Addressing this could lead to a more nuanced and robust explanation model.

---

> ### Author Response · Authors · 2025-03-24
> **Thank you for your comments. We believe we can easily address them.**
>
> **Requested Changes**
>
> *Reviewer: "I suggest that the authors clarify their assumptions regarding the tags and discuss potential practical methods for generating these tags."*
>
> **Response:** We agree that tag creation is important. Our three formulations are relaxed in that they naturally trade-off explaining all instances with explanation complexity. As such they do not have any strong assumptions beyond the tags are typically representative of the images and that tags can contain noise/errors. We will be sure to explain this in the introduction.
>
> In real applications where explanations are critical it could be possible to generate tags using any number of methods such as discussed below.
>
> | **Modality** | **Tag Creation Methods** |
> | --- | --- |
> | People in Images | Commercial software such as Google Photos can tag images with the people within them. The associated files can be downloaded for analysis. For example EXIF files list the Tags/DateTimeOriginal used in Google Photos and IPTC files list the Tags/Caption-Abstract used for the Description in Google Photos |
> | Objects in Images | Many image taggers exist such as blip2, kohya, wd14-swinv2-v2. There are even tag editors \\url{<https://github.com/starik222/BooruDatasetTagManager}>. The creation of StableDiffusion methods for auto-captioning of images also provides methods to generate tags |
> | Emails, articles, documents | LLM in principle have allowed the possibility of tagging documents with key words but the concern was how useful these tags would be on specific domains. However, recent work has shown how to take general LLM and make them task specific taggers of documents, emails etc. See *Tag-LLM: Repurposing General-Purpose LLMs for Specialized Domains ICML 2022* |
> | Graphs | The ability to tag social network users has been in practice for well over a decade and much meta-data on users is captured and available. *Survey on social tagging techniques, SIGKDD 2010, Volume 12, Issue 1*|
>
> *Reviewer: "Additionally, please comment on possible future work addressing the varying degrees of contribution from different tags."*
>
> **Response:** Thanks for this interesting suggestion that generalizes our model. To formulate problems based on this suggestion, we need to introduce edge weights (that indicate how well a tag explains a normal instance or an outlier) and reformulate the problems in terms of these weights. We note that the NP-hardness result for MDMP-WC (Section 4.3) will still hold (since that result only needs each weight to be 1, that is, the contributions of all tags are equal). The complexity aspects of the other problems need to be reconsidered. We will mention this suggestion as a topic for future work.

---

### Review · Reviewer_iVQ3 · 2025-03-17

**Summary Of Contributions:**

The given paper presents an interesting post-hoc explanation technique for Explainable Anomaly Detection. Given a set of outlier data points, normal data points, and an expansive set of feature tags, the technique aims to select feature tags that contrastively explain outlier data points from the normal data points.

The authors formulate the problem in 3 ways, each of which expands upon the previous one in terms of constraints or explanation diversity.

1. The first formulation "Contrastive Explanation" aims to select tags that describe the outlier points such that the number of normal data points described by the tag are less than the number of outlier points. They expanded this formulation by imposing a constraint of selecting only top k tags. This formulation makes sure to select tags/features that identify the points which are more important in describing the outlier points than normal points.

2. The second formulation adds an upper bound on the number of times a feature tag was used by a normal point. This constraint enforces selection of tags that are much more heavily used when explaining outlier points than normal points.

3. The third formulation extends the previous formulation by selecting n different sets of k tags instead of just using 1 set of k tags. This is a nice approach since multiple different outlier groups do occur.

## Analysis of results

The authors provide theoretical analysis for each formulation, determining computational complexity and proposing algorithms for solutions. They evaluate their approach on three datasets spanning different modalities: CelebA (images), COMPAS (tabular data), and HateXplain (text data). The results show that their method successfully identifies meaningful patterns in each domain. For CelebA, the explanations reveal biases in how deep learning models define normality. In COMPAS, the method confirms known biases while providing additional insights about correlations between race and crime types. For HateXplain, it effectively distinguishes hate speech patterns from normal speech. The model-agnostic nature of the approach allows it to be applied to explain results from any anomaly detection algorithm without requiring access to the algorithm's internal mechanisms.

**Audience:**

Yes

**Broader Impact Concerns:**

Given that the paper discusses generation of explanations for anomalies in datasets, it should be kept in mind that explanations, if not correct, and if used for real-world applications can be misleading and may result in cascading issues. Incorrect or imprecise explanations could lead practitioners to focus on the wrong features or patterns when investigating anomalies, hence potentially missing the true causes or wrongly flagging normal instances. This is particularly concerning in high-stakes domains like fraud detection, security monitoring, or healthcare, where decisions based on these explanations could have significant consequences.

**Claims And Evidence:**

Yes

**Requested Changes:**

1. [Recommendation] The authors should try to add metrics for measuring consistency across similar instances. Reliable explanations should identify similar tag sets for anomalies that share key characteristics. One way is to measure variance in tag importance scores across similar instances.

2. [Recommendation] It will be nice to see the sensitivity of the method to key hyperparameters, such as the weight constraint in the second formulation and the number of explanation sets in the third formulation.

**Strengths And Weaknesses:**

## Strengths

1. The authors provide solid theoretical analysis for their formulations, including proofs of computational complexity and efficient algorithms for two of the three formulations.

2. The approach works with any anomaly detection algorithm without requiring access to internal mechanisms, making it widely applicable across different contexts.

3.  The method was validated across three 3 data modalities (images, text, and tabular data), which demonstrates its versatility

## Weaknesses

1. The method relies on pre-existing tags/features. This is a drawback since majority of the dataset are not available with tags which limits the adoption of the approach.

2. The paper primarily relies on qualitative assessment of explanations by checking alignment with known ground truths. However, it lacks quantitative metrics to objectively measure explanation quality. The authors could have included metrics such as consistency across similar instances. Without such quantitative evaluation, it's difficult to compare this approach with future methods or understand the trade-offs between the three proposed formulations.

---

> ### Author Response · Authors · 2025-03-24
> **Thank you for the excellent suggestions. We believe we can address them.**
>
> The reviewer notes that *“The method relies on pre-existing tags/features. This is a drawback since majority of the dataset are not available with tags which limits the adoption of the approach.”*
>
> **Response:** Another reviewer had this concern and we prepared the following table. It shows tags can be generated for many data types due to recent work in foundational ML models.
>
> | **Modality** | **Tag Creation Methods** |
> | --- | --- |
> | People in Images | Commercial software such as Google Photos can tag images with the people within them. The associated files can be downloaded for analysis. For example EXIF files list the Tags/DateTimeOriginal used in Google Photos and IPTC files list the Tags/Caption-Abstract used for the Description in Google Photos |
> | Objects in Images | Many image taggers exist such as blip2, kohya, wd14-swinv2-v2. There are even tag editors \\url{<https://github.com/starik222/BooruDatasetTagManager}>. The creation of StableDiffusion methods for auto-captioning of images also provides methods to generate tags |
> | Emails, articles, documents | LLM in principle have allowed the possibility of tagging documents with key words but the concern was how useful these tags would be on specific domains. However, recent work has shown how to take general LLM and make them task specific taggers of documents, emails etc. See *Tag-LLM: Repurposing General-Purpose LLMs for Specialized Domains ICML 2022* |
> | Graphs | The ability to tag social network users has been in practice for well over a decade and much meta-data on users is captured and available. *Survey on social tagging techniques, SIGKDD 2010, Volume 12, Issue 1*|
>
> **[Recommendation]** *"The authors should try to add metrics for measuring consistency across similar instances. Reliable explanations should identify similar tag sets for anomalies that share key characteristics. One way is to measure variance in tag importance scores across similar instances."*
>
> **Response:** An ongoing challenge with validating XAI methods is how to evaluate explanation quality.  The most authoritative work on the topic [Miller 2019] points out that humans' expectations for explanation are not only complex but varying depending on the person/audience. Hence we believe comparing against known ground truths to be the most convincing approach as novel insights in a new domain add an extra burden of proof that can vary depending on the person.
>
> The reviewer suggests that *“Reliable explanations should identify similar tag sets for anomalies that share key characteristics.”* We believe this is precisely what formulation #3 does. To refresh the reviewer’s mind, our third method is given a collection of outliers and intelligently divides them into $k$ types of outliers which will have similar explanations. In that way it can be viewed as simultaneously finding collections of outliers along with their similar/reliable explanations. For example, Table 3 finds three different types of outliers and explains each: the first type overwhelmingly have black hair, the second type, are wearing hats and the third type have eyeglasses.
>
> The reviewer’s suggestion of consistency across similar instances is an interesting one. It suggests that a high quality explanation is a stable explanation. We can do this easily enough but it seems this is more a measure of the tagging quality not the explanation quality. For example in the CelebA dataset similar looking people will only have similar tags if the taggers were consistent. But perhaps we have misunderstood the suggestion. Please clarify if we have, thank you.
> The reviewer also mentions earlier how to compare our three formulations. We tried to compare our three methods by showing how their coverage differs compared to the population tag statistics. For example Tables 1,2 and 3 compare the coverage for our three formulations on the CelebA data set. If the reviewer has other suggestions then please let us know.
>
> **[Recommendation]** *"It will be nice to see the sensitivity of the method to key hyperparameters, such as the weight constraint in the second formulation and the number of explanation sets in the third formulation."*
>
> **Response:** Thanks for these suggestions, we did explore the weight constraint and the number of explanation sets in the third formulation as follows. For the former, increasing/decreasing weight is just a way to change the complexity of the explanation. We can specifically talk to this in the paper as it’s a good insight we did not discuss. For the latter, we found that more numerous knapsacks/explanations just took an existing explanation and divided it into two. For example, Table 3 has three explanations/knapsacks, changing this to four just takes the last explanation and breaks it into one group who are predominantly black and another that have eye glasses so it wasn’t particularly interesting to put into the paper but we can add this in as the reviewer believes it is important.

---

### Review · Reviewer_WKkh · 2025-03-17

**Summary Of Contributions:**

As repeated in abstract, introduction and conclusion, this paper:
- deals with the problem of Explainable Anomaly detection (XAD).
- introduces an original bridge between tag subset slection and the Knapsack problem
- it is data-based and not model-based: it deals with tags in the data to explain how some ML model made its decision on classifying which are the normal/anomalous examples.
- it thus deals with defining what is normal too.
- provides time complexity proofs and algorithms.

**Audience:**

Yes

**Broader Impact Concerns:**

AD has impacts, but precisely this paper contributes to providing explanations for AD ML models, so it's a priori positive.

A downside note: having good and well-grouned algorithms to explain the decisions of models seems good but may induce a kind of ``The Mathiness Effect'' (or Formalism Bias or "Mathematical Seduction" effects): one would trust the explainer outputs because it is rigorous, but explanations may still be wrong (if e.g. some tags are missing and the ML, decision-making-model used some features that are not covered by the tags).
Maybe this should be quickly mentionned somewhere (as a limitaiton of the model).

**Claims And Evidence:**

Yes

**Requested Changes:**

I divide my list of changes in required vs suggested, sorted more or less by order of importance.
Note: I was trained as a Physicist, with an applied maths background. I am not too much into formal proofs, more about intuitions. This is my perspective. It may not be representative of the whle community.


# Required changes

## 1. Observation 3.1, coverage vs edge count:

Very interestingly, authors state that
> Computations based on coverage typically are computationally intractable as they can model versions of the minimum set cover problem (Garey & Johnson, 1979).

I think the paragraph preceding Observation 3.1 is confuising. In particular, the part where p(y), w(y) are interpreted as number of instances covered by y :
> For any tag node y, the profit of y, denoted by π(y), is the number of outlier instances covered by y; that is, p(y) = |O(y)|. The weight of any tag y, denoted by w(y), is the number of normal instances covered by y; that is, w(y) = |N(y)|. For any tag node y, the utility of y, denoted by µ(y), is given by p(y) − w(y); thus, the utility of a tag is the number of outlier instances covered by y minus the number of normal instances covered by y.

This is correct but somehow suggests that p(Y) and w(Y) also reflect this coverage, when it does not. I initially thought this was a mathematical error, until I read again the paragraph, noticing that it was discussing w(y) and not w(Y). The coverage is equal to the edge count only for single-element set Y={y}.

I think a small remark, right before or right after observation 3.1, about the fact that the edge count is always larger or equal to the number of covered instances, is needed.
I note that the weight is an upper bound to the number of normal instances covered (fine since we minimize it), but the profit is also an upper bound to the number of anomalies covered (not nice since we try to maximize it). There could be profite-monster-tags (as in utility monster) that are nasty for your algorithms.


## 2. Claim about tags being used for first time ?

There is a strong claim:
> As will be pointed out in in the related work section (i.e., Section 6), existing work for XAD does not use semantic tags, generate contrastive explanation or provide a comparable global level explanation; hence, there are no direct comparisons to be made.

However, it is not very clear why (semantic) tags are that different from features lying in some range. Any method that usees features' values for explanations could readily be used on tags, when available, no ?

Please discuss that carefully, or, diminish the claim. In any case, the connection and differences need to be made very clear to the reader.

Similary, for SHAP values, which are entirely absent from the text and the citations' titles. Using them as tags seems pretty straightforward. I believe the connection/distinction with your approach needs to be made clear.

Along the same line, it is later written:
> In contrast, our work is model level and uses semantic tags as a mechanism to generate explanations so does not rely on the feature space being interpretable.

But, if you do not have semantic tags (interpretable features attached to the data or produced by some algorithm, such as CLIP, although they may not be used for the classification/AD), then you cannot apply your algorith ? So in this sense you are not doing better than explability models which need interpretable tags, right ?
This sentence is quite misleading.

## 3. A note about experiments.

In *A note about experiments.*, you normalize Outliers and Normal samples separately.
I believe this is not a neutral point at all, and that using instead
$$(E^Y_\mathbb{O} - E^Y_\mathbb{N}) / |E_\mathbb{O}+ E_\mathbb{N}|$$
as normalization, would allow to compare various sizes of datasets, but change the result considerably. In practice, anomalies are usually rare, and if in practice, $ (E^Y_\mathbb{O} - E^Y_\mathbb{N})$ is often negative, that's an intersting output.
In practice your definition corresponds to considering a
$(E^Y_\mathbb{O} - \rho E^Y_\mathbb{N})/const.$, where $\rho$ is related to a measure of the class imbalance, namely $E^Y_\mathbb{O} / E^Y_\mathbb{N}$.

1. I think it would be worth discussing this.

2. Ideally, discuss some results with the other normalization that I suggest, or some variant of it.


## 3. (non)sorting - time complexity
> Thus, we can find the top-k tags with respect to utility in O(|V T |) time.

I believe a factor $k$  is missing. Altough $k$  may be small, I find it very confusing to not make it appear in the time complexity computation.
When there are many tags, $k$ may actually become rather large.




## 4. Notations

### 1. p and \pi
Around *Optimization Objectives.* the notation $p(y), p(Y)$ is introduced for the profit associated to y, Y. Then it becomes $\pi(y),\pi(Y)$. This is confusing and inconsistent. Please stick to $\pi$ (also through algorithms)

### 2. Knapsack problem vs rest of paper:
Here it may be my education as a Physicist speaking (not caring too much about abuse of notations) but in the section *Knapsack Problem* you introduce $a_i$ as objects (that are then corresponding to tags). Why not denote them as $u_i$ ? Or $y_i$ actually (see next point).

### 3. Tags

Tags are initially denoted y (for a single one) or $y_i$ for tage number $i$. Why shift to $u_i$ after (and yet keep using $Y$ to denote a subset of $\mathbb{U}$). You could use $y_i, Y, \mathbb{Y}$ and remain consistent. This would made reading even easier.

### 4. Group #1

In table 3, you write group #1, 2, 3.
This kind of implies outlier groups, when actually it's tag groups. (I noticed this because In Table 3, I noticed that Black_hair is 36% pf the outlier group. Same in table 2.)
Maybe you could write "Tags Group #1" ?
Same in table 8, 11.




## 5. Explanation

> However, the version of Knapsack arising in the context of contrastive explanation involves numbers whose values are bounded by polynomial functions of the problem size.

Here it's better to state briefly why, intuitively, rather than waiting towards the end of the paper.
In the end it's simply that the maximum weight or profit for a given tag is at worst/best, the number of instances. It does not take long to say that.




## 6. Illustrative Example Using MNIST

In the paragraph *Illustrative Example Using MNIST*,
> For each of the ten digits, we created a different outlier detection problem and generated an explanation for each.
> (...)
> We apply a basic auto-encoder to determine outliers based on their reconstruction error for each digit class separately.

1. These sentences were very unclear to me at first, in the sense that I did not understand the experimental setup. I then saw that it is a rather conventional setup for AD, but, it would not take long to extend the sentence into a couple of them to clarify.

As I understand it:
- You train an AE to reconstruct images of the train set, and then consider (for the test set?) as anomalies the top 5% least well reconstructed images, and as normal the other 95%. This setup is considered as the AD ML model.
- Then your XAD tries to explain the decisions of this ML model (or rather, using test data, explain what is the definition of normality/anomaly).
- And, you repeat this for the 10 digits independently, meaning that you have 10 independent AE models, each trained to reconstruct only one class of digit ? (this is the part where I'm still not clear). Why not use a single AE for all digits?

2. Also, you say *basic autoencoder*. You could be a bit more precise ? 2 layers MLP with a bottleneck, or something with some convolution layers ?


## 7. Conclusion of sec 4.3

I am troubled by the conlusion of sec 4.3. Honestly, this is probably the part that is most technically challenging for me, so I may have misunderstood some things.

It's not clear to me why constructing (like you do in sec 4.3), for ANY 3-parition setup, AN instance of the MDPC-WC problem that matches it, proves that ANY MDPC-WC problem can be mapped to A 3-partition? Or bounded in complexity by another 3-partition problem ? Which would prove the complexity result.

I have not tried to find one, but it feels like there could exist MDPC-WC instance that do not map to any 3-partition problem.













# Suggested changes

Here I list suggestions that are not required.

## 8. Other directions, discussions

In formulation 3, the requirement that the explanations are non-overlapping (sets of tags are distinct) is a strong one. I can think of cases where distinct groups of outliers would be epxlain by distinct explanations, but, with some overlapping tags. (like, being female: still, many groups are sub-groups of "females").


## 9. Still on coverage:

> Computations based on coverage typically are computationally intractable as they can model versions of the minimum set cover problem (Garey & Johnson, 1979).

A quick discussion about how using coverage as a metric (which sounds better than edge count to produce good explanations) would be very nice, maybe in the conclusion, as direction for future work ?

## 10. Clarity

page 11, point 2.
> Thus, each normal and outlier instance is associated with exactly one tag.

I found point 2., around this sentence, not very clear. You could extend to be clearer, maybe.





## 11. Repetitions, (lack of) conciseness

### sec 2 and sec 3.
I enjoyed the overall clarity, and in particular section 2.
However, when reading section 3, it was not adding much to sec 2, w.r.t. the length of sec 3.
I would either shorten a lot the text of sec. 2 to remain very high level and not defining everything (not my favorite option); or, better, simply merge section 3 into sec 2. My point is that very little new concepts or notations are introduced in sec. 3: when I read sec 2, I consulted figure 1, and then learnt basically nithing reading sec 3.

### Inside sec 3.

Between each **Required** paragraph and the paragraph right after it, I see no added value, it's just that the meaning of some quantities is recalled in words.


### Knapsack problem

In *Required*, why not use maths notations s.a. $argmax()_{Y\in \mathbb{U}, constraint}$ ?
Also, again, a good example of useless sentence:

> Our next lemma points out that DMP-WC problem is indeed an instance of the Knapsack problem.
> Lemma 4.2. The DMP-WC problem is an instance of the Knapsack problem.

-> clearly the 1st line is useless.





## 12. Typos:
- ...are connected to only to the tags...
- the goal is find a subset  -> the goal is TO find a subset
- algorithm 2, line 7 : w(p_i) -> w(u_i)  (that I would denote w(y_i) to remain consistent)

**Strengths And Weaknesses:**

Strengths:
- original bridge between fields
- clearly written
- pedagogic, with a first algo which is trivial, up to the 3rd, quite non trivial
- experiments on 3 datasets with very different modalities

Weaknesses:
- a bit lengthy, at times.
- some claims slightly over-stated / not discussed carefully enough
- lack of comparison with previous litterature (might not be possible to address). Although the approach is original, one may be able to better discuss link with previous research using features and not tags (see point 2. "Claim about tags being used for first time ?" below)
- some more less important "details", mentionned below (see requested changes)

---

> ### Author Response · Authors · 2025-03-26
> **Thank you for your detailed comments. We have addressed all the required changes**
>
> We thank the reviewer for their insights. Given their acknowledgement of their background and unfamiliarity with theoretical CS, we responded accordingly and apologize if our responses appear to be too simplistic!
>
> *“1. Observation 3.1, coverage vs edge count:"*
>
> **Response:** The reviewer is correct, we did interchangeably use \pi(y) and p(y). Going forward we will use just \pi(y).  Also, we will add a note as suggested to point out that for a subset Y of tags, the value of \pi(Y) is at least as large as the number of outliers covered by the set Y and that it is an upper bound on the number of outliers covered.
>
> *"2. Claim about tags being used for first time ?"*
>
> **Response:** Whilst we did say features could be used as tags, a far more common use of our work is tags are generated by human annotation or by using foundational models (this is how the CelebA tags were generated) so are quite different from the reviewer's examples using SHAP explanations. We produced a Table given to the other reviewers to show how tags can be automatically generated for many modalities of data using foundational models. We can't include it here again due to space limitations so please see our other replies
>
>  *"3. A note about experiments."*
>
> **Response:**  We have used the normalization $E_O^Y/|E_O| −E_N^Y/|E_N|$ and the reviewer suggests using $(E_O^Y−E_N^Y)/|E_O+E_N|$.
>
> As mentioned in the paper “we normalize values of parameters by constants to make the computational results more semantically meaningful and comparable across datasets of different sizes”.
> Our normalization scheme reports the difference between two probabilities (i.e. fraction of all POSSIBLE edges covered by the explanation). Looking at Figure 1 it is the fraction of edges (as given by the outlier explanation) incident on the outliers less the fraction of edges (as given by the normal explanation) incident on the normal points. The larger probabilty the better the explanation.
>
> The reviewer’s suggestion seems interesting but we don’t see how it is more interpretable as it produces a single number that can’t be interpreted as a probability.
>
> *“3. (non)sorting - time complexity"*
>
> **Response:** The algorithm that our paper refers to is a linear time selection algorithm ( Section 9.3 of the textbook “Cormen et al. 2009” that we cite) and the time complexity is **for all $k$ values.** In particular, the constant hidden by the big-O notation for the running time does not depend on $k$.
>
> *"4. Notations 2. Knapsack problem ..."*
>
> **Response:** As is standard practice in the field, we define a general version of the knapsack problem using the notation/parameters a_i and q(a_i) but then define a specific instance of it for explanation and introduce new notation/parameters and the correspondence between the parameters of the original problem we are reducing from. This is to show a simple reduction of knapsack to the XAI problem we study and again is common in the field to generate a polynomial time algorithm.
>
> *"4. Notations 4. Group #1 In table 3, you write group #1, 2, 3. This kind of implies outlier groups, when actually it's tag groups."*
>
> **Response:** Our method simultaneously divides all outliers into $k$ groups (in this case $k=3$) whilst generating an explanation for each group. There is no overlap between the groups. So we think calling them outlier groups makes sense as they are explanations for different groups of outliers.
>
> *"5. Explanation"*
>
> **Response:**  As suggested by the reviewer, we will mention that the efficient solvability of the Knapsack problem that arises in the context of the DMP-WC problem is due to the fact that the maximum values of profits and weights are bounded by polynomials in the input size.
>
> *"6. Illustrative Example Using MNIST"*
>
> **Response:** The reviewer's understanding of the experiment is correct except there is no test set here as it’s unsupervised. The top 5% of least well reconstructed images in the data set used to build the AE are the outliers.
> Further, by basic AE we mean the standard AE typically used with this simple data set: one fully connected layer for the encoder and decoder. There is no need for multiple layers or convolutions. We will make this clearer so there is no uncertainty.
>
> *"7. Conclusion of sec 4.3"*
>
> **Response:**  There seem to be some simple misunderstandings that we try to explain below. To prove the NP-completeness of the MDMP-WC problem (Section 4.3), the standard approach is to show that the problem is in the class NP and that a known NP-complete problem (which in our case is 3-Partition) can be reduced to MDMP-WC. That is exactly what we have done in Section 4.3. The existence of some MDMP-WC instances that may not be reducible to 3-Partition doesn’t affect the validity of the result.
>
> **Suggested changes by reviewer**
>
> Response: We thank the reviewer for their suggested changes. We will try to incorporate as many of them as possible given the time limitations.

---

> > ### Comment · Reviewer_WKkh · 2025-04-01
> > **Quick answer**
> >
> > I thank the authors for their answers. I am happy with most answers and promises for changes, here are a few additional points:
> >
> > ### About tag generation:
> >
> > Obviously this is a crucial point (was raised by the 3 referees). It's quite central to the manuscript, and determines the scope of the method.
> >
> > The answer on tags needs to be assorted with a well-written change in the manuscript.
> >
> > The fact that foundation models offer the possibility to tag data is nice, however the results of the proposed XAD method would then rely on the quality of the tagging from these foundation models. This is a limitation that should be outlined.
> > To be provocative: How can we explain the foundation models ?
> >
> >
> > ### About tag groups vs outlier groups ("4. Notations 4"):
> >
> > Thank you for your answer. But then, the percentages shown are a bit misleading, because the denominator used still covers the whole (outlier) population, and not the (outlier) population within the group. I think that re-calculating these percentages to show how much tag 1 is prevalent in group 1 (e.g. Black_Hair for Group #1) will actually be beneficial to supporting the claim of the paper: the percentage will increase from 36% to.. more.
> >
> > An alternative or complementary possibility would be, to show the % that group 1 represents within the (complete) population. Then one can compare with the population coverage.
> >
> > ### 4. Notations 2. Knapsack problem ..."
> >
> > Ok, I understand. Then, please try to restrict the use of new notations to the original Knapsack problem and the mapping with XAD. When you discuss XAD, try to maintain the same set of notations throughout the paper, and algorithms.
> >
> > For instance in algorithm 2 you use $u_i\in \mathbb{U}$ for tags instead of $y\in \mathbb{Y}$, as done previously (above observation 3.1 for instance).
> >
> > ### Other point
> >
> > Referee GHZM pointed out, rightly so, that:
> > > While exploring such a relationship can indeed offer insights into Deep SVDD's behavior, it relies heavily on the assumption that the tags are strongly correlated with the predictive input features. The authors should explicitly state this assumption and clarify that their method depends on predefined, high-quality tags closely associated with these predictive features.
> >
> > This is an important point, that is somehow discussed in the manuscript, but not in enough detail.
> >
> > Indeed, the proposed XAD method explains model decisions instance by instance, not basing its explanations on model weights. It's more a post-hoc measure of biases of the model explained, than an explanation of how this model actually make decisions.
> >
> > The empirically measured biases can be considered a good proxy of the intrinsic model biases, esp. in the limit of large number of samples. However, (as stated in the paper), it does not necessarily reflects how the model actual makes decisions.
> > This should be recalled in the conclusions, quite explicitly.

---

> ### Author Response · Authors · 2025-04-01
> **We are glad you liked our responses and can address your new suggestions**
>
> *"About tag generation
> Obviously this is a crucial point (was raised by the 3 referees). It's quite central to the
> manuscript, and determines the scope of the method. The answer on tags needs to be
> assorted with a well-written change in the manuscript. ..."*
>
> **Response:** This is a very good suggestion. We will create a new sub-section (“Tag Generation, Assumptions and Limitations”)
>  that brings together all issues related to tags including those currently spread through out the paper.
>
> This will include: i) sources of tags, ii) that our method can accept tags that can contain some noise but iii) makes the
>  assumption that the tags are good enough to be correlated with the algorithm’s actions. We will also point out that a contrastive
>  explanation can naturally identify if the tags are not correlated well with the algorithm outcomes. Namely that there does not
> exist a  strongly contrastive explanation. For example in Table 1 if the two percentages reported for each tag found in the
> explanation were approximately the same (or not that different) that would indicate a non-constrastive (and poor) explanation.
>
> Finally, we will mention that like just about all XAI methods (to our knowledge), our method does not explain the thought process
>  of the deep learner (that topic is known as *meta-cogntion* in the cognitive science literature) and we will provide a few
> citations to that.
>
> *"About tag groups vs outlier groups ("4. Notations 4")
> Thank you for your answer. But then, the percentages shown are a bit misleading, because the denominator used still covers the
>  whole (outlier) population, and not the (outlier) population within the group."*
>
> **Response:** Apologies, that the headings (in Table 3 for example) are not clearer. The percentages are exactly what the
>  reviewer thinks they should be. For example, 36% of the outliers *in group 1* have Black_Hair and just 2% of the normal
>  assigned individuals (also in group 1)  have Black_Hair. We can make this clearer by adding to the caption.
>
> *"4. Notations 2. Knapsack problem ... Ok, I understand. Then, please try to restrict the use of new notations to the original Knapsack problem and the mapping with XAD."*
>
> **Response:** Thank you we can maintain the notation for the Knapsack problem and will clearly point out
> that for the reduction to show a polynomial time algorithm we need to introduce new notation (for the XAD problem) as is standard in the theory literature.
>
> *"Other point
> Referee GHZM pointed out, rightly so, that:
> While exploring such a relationship can indeed offer insights into Deep SVDD's behavior, it relies heavily on the assumption that
>  the tags are strongly correlated with the predictive input features. The authors should explicitly state this assumption and clarify
>  that their method depends on predefined, high-quality tags closely associated with these predictive features."*
>
> **Response:** We agree that these are important points. See earlier response to your first point. We will include answers to these
>  points in the new section on ““Tag Generation, Assumptions and Limitations”.
>
> We will also comment, though tag quality is important, it is not a too hard problem to get quality tags. Our experiments showed it was possible to get very useful and intuitive explanations from tags for three different settings: CelebA (Tags from human annotators and foundational models), Compas (Tags from database records) and HateExplain (Tags from human language).

---

### Decision · Action_Editor_x5TX · 2025-05-02

**Recommendation:** Accept as is

**Comment:**

This paper presents a contrastive explanation for anomaly detection. I recommend authors to make the code public for reproducibility.

**Audience:**

This paper focuses on explainable anomaly detection. The tackled problem is important in decision making and is highly relevant for the TMLR audience.

**Claims And Evidence:**

Initially some claims such as being first to explore tags were overstated. Reviewer iVQ3 also mentioned to include more evaluations such as human evaluation. After long discussions the authors addressed most of the issues brought up by reviewers, softened the claims, and promised to discuss limitations of the proposed method in more detail. They also committed to make the notation more consistent throughout the manuscript. All reviewers agreed on acceptance of the work. I also lean towards acceptance.